# Retrieval-Augmented Language Model for Knowledge-Aware Protein Encoding

**Jiasheng Zhang** [1]  **Delvin Ce Zhang** [2]  **Shuang Liang** [1]  **Zhengpin Li** [3]  **Rex Ying** [4]  **Jie Shao** [1]

## Abstract

Protein language models often struggle to capture biological functions due to their lack of factual knowledge (*e.g.,* gene descriptions). Existing solutions leverage protein knowledge graphs (PKGs) as auxiliary pre-training objectives, but lack explicit integration of task-oriented knowledge, making them suffer from limited knowledge exploitation and catastrophic forgetting. The root cause is that they fail to align PKGs with task-specific data, forcing their knowledge modeling to adapt to the knowledge-isolated nature of downstream tasks. In this paper, we propose **K**nowledge-**a**ware **r**etrieval **a**ugmented protein language model (**Kara**), achieving the first task-oriented and explicit integration of PKGs and protein language models. With a knowledge retriever learning to predict linkages between PKG and task proteins, Kara unifies the knowledge integration of the pre-training and fine-tuning stages with a structure-based regularization, mitigating catastrophic forgetting. To ensure task-oriented integration, Kara uses contextualized virtual tokens to extract graph context as task-specific knowledge for new proteins. Experiments show that Kara outperforms existing knowledge-enhanced models in 6 representative tasks, achieving on average 5.1% improvements.

## 1. Introduction

Proteins are essential for understanding biological processes and recent advances in artificial intelligence led to growing interest in learning generalized vector representations of proteins (Hu et al., 2024). By viewing amino acids as language tokens, protein language models (PLMs) such as ESM (Lin et al., 2023), ProteinBert (Brandes et al., 2022), and ProtBert (Ahmed et al., 2022) have proven highly valuable in various application tasks such as drug discovery (Hoang et al., 2024) and function prediction (Xu et al., 2024; Shaw et al., 2024). However, as pointed out by Kalifa et al. (2024); Zhou et al. (2023); Zhang et al. (2022a), lacking factual knowledge (*e.g.,* gene descriptions) makes them struggle to capture intricate biological function encoded within protein sequences.

Existing solutions leverage protein knowledge graphs (PKGs) that describe the relationships between proteins and gene ontology (GO) entities with biological relations (Chen et al., 2023b). These models use protein sequences and associated GO annotations as complementary encoding objectives during pre-training. For example, OntoProtein (Zhang et al., 2022a) uses the TransE objective (Bordes et al., 2013) to optimize the alignment between protein representations and associated GO entity representations. KeAP (Zhou et al., 2023) uses GO entity representations to guide masked token prediction of protein sequences via a cross-attention mechanism. Despite their effectiveness, unfortunately, they lack explicit integration of task-oriented knowledge.

**Limitations. 1) Implicitly embed knowledge information.** Existing methods use knowledge only as encoding objectives to supervise the pre-training of the model, assuming that knowledge information can be well embedded within model parameters. However, as highlighted by Kandpal et al. (2023), LMs often struggle to precisely embed knowledge, particularly long-tail knowledge. Storing knowledge within model parameters also makes them unable to adapt to knowledge graph updates (e.g., adding new knowledge), which further diminishes their usability. **2) Overlook the structure information.** Existing methods treat each knowledge triplet (*i.e.,* $(protein, relation, GO)$) independently. However, the neighboring GO entities of a protein are often correlated, and the high-order connections between proteins (*e.g.,* proteins linked to a GO entity through similar relations) can provide additional insights into their functional similarities. Ignoring the structural relevance makes existing methods fail to fully exploit knowledge information within PKGs. **3) Lack of task-oriented knowledge modeling.** Existing methods are unable to incorporate knowledge modeling during task fine-tuning, leading to inconsistent optimization objectives between the pre-training and fine-tuning stages. This inconsistency can cause the knowledge learned during pre-training to be catastrophically forgotten when applied to downstream tasks (Lee et al., 2020), while also making the

---

[1]University of Electronic Science and Technology of China [2]The Pennsylvania State University [3]Fudan University [4]Yale University. Correspondence to: Jie Shao <shaojie@uestc.edu.cn>.

*Proceedings of the $42^{nd}$ International Conference on Machine Learning*, Vancouver, Canada. PMLR 267, 2025.

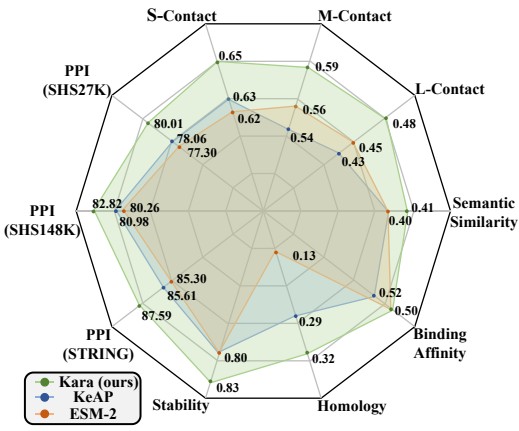

*Figure 1.* Performance in downstream tasks. S-, M-, and L-Contact are the short-range, medium-range, and long-range contact prediction. PPI is the protein-protein interaction prediction.

knowledge extraction process lack task-oriented adjustment.

The root cause is that proteins in downstream tasks often fall outside the PKG, restraining the use of knowledge during fine-tuning. Existing methods fail to align data from downstream tasks with PKGs, forcing their knowledge modeling to adapt to the knowledge-isolated nature of these tasks.

**Proposed Work.** To tackle these limitations, we propose **Kara**, a **K**nowledge-**a**ware **r**etrieval-**a**ugmented protein language model, achieving the first unified and direct integration of PKGs and protein language models. As the core of Kara, we propose a knowledge retriever that can accurately predict potential gene descriptions for new proteins and thus align them with PKGs. This alignment allows the pre-training and fine-tuning stages of Kara to be enhanced through a unified knowledge modeling process, and seamlessly adapt to knowledge updates. By employing contextualized virtual tokens, we achieve token-level information fusion between protein sequence and knowledge. Specifically, we categorized the virtual tokens into knowledge tokens and structure tokens, enabling the explicit injection of high-order graph context as task-oriented knowledge. To unify the optimization objectives, we incorporate structure-based regularization into both two stages, bringing function similarities into protein representations and helping the pre-trained knowledge to be effectively transferred to downstream tasks.

As shown in Figure 1, experiments in 6 representative tasks show the effectiveness of Kara. It outperforms powerful baselines (*i.e.,* KeAP and ESM-2) across all the tasks. For instance, Kara exceeds the state-of-the-art knowledge-enhanced model KeAP by $11.6\%$ in the long-range contact prediction and by $10.3\%$ in the protein homology detection, highlighting Kara as a better paradigm for integrating protein knowledge graphs into protein language models.

## 2. Preliminaries

**Protein Knowledge Graph.** A protein knowledge graph (PKG) is $G = \{V_p, V_{go}, R, F\}$, where $V_p$ is the protein set and $V_{go}$ is the gene ontology (GO) entity set. $R$ is the set of relations among proteins and GO entities. The knowledge set $F$ consists of two kinds of triplets: $(p,r,g)$ which describes the properties of proteins, and $(g_1,r,g_2)$ which describes the relationships between GO entities. Each protein $p \in V_p$ has an amino acid sequence $s$. Each GO entity $g \in V_{go}$ includes a text description $t_g$ explaining the gene's function. Similarly, each relation $r \in R$ comes with a text description $t_r$. We first generate pre-trained embeddings of items in PKG and store them in vector databases for further usage. Specifically, relation $r$ and GO entity $g$ are encoded based on their text descriptions using a frozen PubMedBERT model (Gu et al., 2021), resulting in relation embedding $\mathbf{r}$ and GO embedding $\mathbf{g}$. Protein $p$ is encoded based on its amino acid sequence via a frozen ProtBert model (Ahmed et al., 2022), resulting in protein embedding $\mathbf{p}$. These stored embeddings will be further used to construct virtual tokens in Kara. As in previous works, we use the ProteinKG25 knowledge graph (Zhang et al., 2022a). Detailed introduction of ProteinKG25 can be found in Appendix B.

**Problem Formulation.** Given a PKG $G$, we aim to pre-train a knowledge-aware protein language model $f$ so that for each protein with amino acid sequence $s$, we generate its knowledge-integrated representation as $\tilde{\mathbf{p}} = f(G,s)$. In Kara, $f$ consists of a protein encoder, a knowledge projector, a protein projector, and a knowledge retriever. We use ProtBert (Ahmed et al., 2022) as the backbone of the protein encoder, the same as previous works (Zhou et al., 2023) for a fair comparison. We will test different backbones in Section 4.6. By fine-tuning $f$ on task-specific data, we further verify its capabilities to generalize pre-trained knowledge to downstream tasks (*e.g.,* protein homology detection).

## 3. Methodologies

As shown in Figure 2, with a knowledge retriever to align new proteins with the protein knowledge graph, Kara can uniformly integrate knowledge information during both the pre-training and fine-tuning stages. Specifically, the contextualized virtual tokens allow Kara to explicitly inject task-oriented knowledge and high-order structure information into protein representations. During pre-training, masked language modeling (MLM) helps the protein encoder learn to fuse the information of protein sequences and structured knowledge at the token level. During fine-tuning, downstream task modeling helps the protein encoder learn to extract task-specific useful knowledge from PKGs via virtual tokens. Additionally, based on the high-order connectivity between proteins, structure-based regularization is incorporated during the two stages to unify their optimization

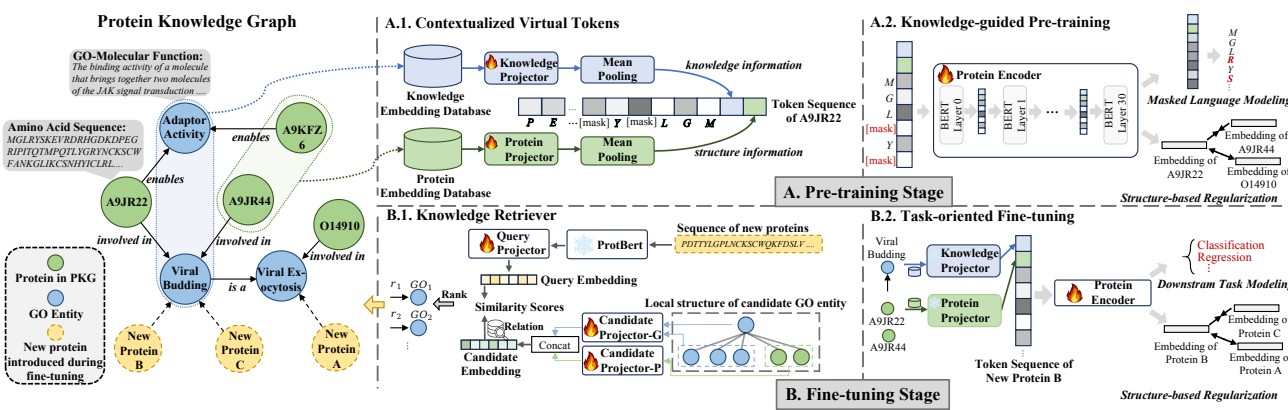

*Figure 2.* **Overall architecture**. During pre-training, Kara directly integrates knowledge information via contextualized virtual tokens and structure-based regularization. During fine-tuning, the knowledge information can be similarly integrated into protein representations through a knowledge retriever, which can align new proteins in downstream tasks with the protein knowledge graph.

objectives and inject function similarities into protein representations. We summarize main notations in Table 11 and provide a detailed analysis of Kara in Appendix C.

## 3.1. Pre-training Stage

### 3.1.1. CONTEXTUALIZED VIRTUAL TOKENS

Existing protein language models struggle to encode knowledge information, since 1) knowledge in PKG is interconnected, providing the context of proteins based on the graph structure, but language models are only designed to encode sequential data, limiting their ability to capture graph information; and 2) PKGs contain multi-modal information (*e.g.,* amino acid sequences and GO text descriptions), and protein language models can only encode amino acid sequences, failing to achieve effective multi-modal information fusion. As shown in Figure 2 A.1, we tackle the above challenges by introducing contextualized virtual tokens. By summarizing the associated knowledge of a protein as knowledge virtual tokens and summarizing its high-order structure as structure virtual tokens, Kara can directly inject the knowledge and graph information into protein representations. These virtual tokens are then concatenated with the amino acid token sequences as the knowledge context, so that each amino acid can query them to integrate helpful knowledge information, enabling effective token-level multi-modal information fusion. Specifically, for each protein $p_i \in V_p$, we extract its one-hop GO entities with relations $\mathcal{N}_1(p_i) = \{(r_i, g_i)|(p_i, r_i, g_i) \in F\}$ as its knowledge, and use its two-hop proteins $\mathcal{N}_2(p_i) = \{p_j|(p_j, r_i, g_i) \in F; (r_i, g_i) \in \mathcal{N}_1(p_i)\}$ as its structure context. The knowledge virtual token of protein $p_i$ is then constructed as

$$\mathbf{v}_i^k = \frac{1}{|\mathcal{N}_1(p_i)|} \sum_{(r_i, g_i) \in \mathcal{N}_1(p_i)} \text{MLP}_{knowledge}([\mathbf{r}_i : \mathbf{g}_i]), \quad (1)$$

where $\mathbf{r}_i$ and $\mathbf{g}_i$ are respectively the pre-trained embeddings of relation $r_i$ and GO entity $g_i$ (see Section 2). $[:]$ is the

concatenation operation. $\text{MLP}_{knowledge}$ is a trainable multi-layer perceptron used to project text-modal information into a uniform semantic space. Similarly, to incorporate structure information of $p_i$, we construct its structure virtual token as

$$\mathbf{v}_i^p = \frac{1}{|\mathcal{N}_2(p_i)|} \sum_{p_j \in \mathcal{N}_2(p_i)} \text{MLP}_{structure}(\mathbf{p}_j), \quad (2)$$

where $\mathbf{p}_j$ is the pre-trained embedding of protein $p_j$. $\text{MLP}_{structure}$ is another trainable multi-layer perceptron used to project the amino acid sequence-modal information. We then construct the input embedding sequence for the protein encoder by concatenating virtual tokens with amino acid tokens. Given the amino acid sequence $s_i = [s_i^1, s_i^2, ..., s_i^{|s_i|}]$ of protein $p_i$, where $s_i^m$ represents an amino acid, we lookup the embedding vocabulary of protein encoder to initialize the input embedding sequence as $\mathbf{S}_i = [\mathbf{s}_i^1, \mathbf{s}_i^2, ..., \mathbf{s}_i^{|s_i|}] \in \mathbb{R}^{|s_i| \times d}$, then concatenate it as

$$\mathbf{S}_i \leftarrow [\mathbf{v}_i^k, \mathbf{v}_i^p, \mathbf{S}_i] \in \mathbb{R}^{(2+|s_i|) \times d}. \quad (3)$$

$|s_i|$ is the length of amino acid sequence $s_i$, and $d$ is embedding dimension. During inference, any related knowledge updates can be perceived by constructing virtual tokens.

### 3.1.2. KNOWLEDGE-GUIDED PRE-TRAINING

The pre-training of Kara has two purposes: 1) achieving effective information fusion of the contextualized virtual tokens (*i.e.,* knowledge and structure information) and the amino acid tokens (*i.e.,* protein information); and 2) integrating the knowledge-based relevance (*i.e.,* function similarities) among proteins into their representations. For the first purpose, we introduce knowledge-guided masked language modeling, allowing each amino acid to query the virtual tokens to extract helpful knowledge information for restoring masked tokens, which achieves token-level information fusion at each layer of the protein encoder. Specifically, given the input embedding sequence $\mathbf{S}_i$, we use 15% probability to mask each amino acid token (*i.e.,* replace the amino acid

embedding with the embedding of special token '[MASK]'). The masked embedding sequence is encoded by the Transformer component (Vaswani et al., 2017) as follows:

$$\tilde{\mathbf{S}}_i^l = \text{LN}(\mathbf{S}_i^l + \text{MHA}(\mathbf{S}_i^l)), \tag{4}$$

$$\mathbf{S}_i^{(l+1)} = \text{LN}(\tilde{\mathbf{S}}_i^l + \text{MLP}(\tilde{\mathbf{S}}_i^l)), \tag{5}$$

where $\mathbf{S}_i^0$ is initiated by $\mathbf{S}_i$. LN is layer-norm unit and MHA denotes the multi-head attention unit. After modeling the correlations among virtual tokens and amino acid tokens layer by layer, we leverage cross-entropy loss $\mathcal{L}_{\text{MLM}}$ on the last-layer token embeddings (i.e., $\mathbf{S}_i^L$, where $L$ is the number of layers in protein encoder) to estimate the masked tokens.

While the aforementioned masked language modeling achieves token-level multi-modal knowledge infusion, we further introduce a sequence-level regularization based on graph connectivity between proteins, integrating biological function similarities into their representations. As we mentioned before, each protein $p_j \in \mathcal{N}_2(p_i)$ is two-hop connected with $p_i$ in graph structure. This high-order connectivity indicates that $p_i$ and $p_j$ share the same knowledge $(r_i, g_i)$ and thus should be similar in their biological functions. Therefore, each pair $(p_i, p_j \in \mathcal{N}_2(p_i))$ can be regarded as positive pair that we hope their embeddings are closer in semantic space (e.g., A9JR22 and A9JR44 in Figure 2), and $(p_i, p_k \notin \mathcal{N}_2(p_i))$ can be regarded as negative pair (e.g., A9JR22 and O14910). Specifically, in Kara, we generate the sequence-level embedding of protein $p_i$ as $\tilde{\mathbf{p}}_i = \text{MEAN}(\mathbf{S}_i^L[2:])$, where MEAN is the mean-pooling operation, and $\mathbf{S}_i^L[2:]$ is the last layer token embeddings except the virtual tokens. Then, we apply the margin loss on sequence-level protein embeddings to ensure high-order connected protein $p_j$ is closer to $p_i$ than other proteins.

$$\mathcal{L}_{\text{reg}} = -\frac{1}{|\mathcal{N}_2(p_i)|} \sum_{p_j \in \mathcal{N}_2(p_i)} \text{MAX}(0, \text{sim}(\tilde{\mathbf{p}}_i, \tilde{\mathbf{p}}_j)) \tag{6}$$
$$-\text{sim}(\tilde{\mathbf{p}}_i, \tilde{\mathbf{p}}_k) + \gamma),$$

where sim indicates the similarity function (e.g., cosine similarity). We finally pre-train the parameters within the protein encoder, knowledge projector, and structure projector by jointly optimizing $\mathcal{L}_{\text{MLM}}$ and $\mathcal{L}_{\text{reg}}$. These three components are then used to handle downstream tasks.

## 3.2. Fine-tuning Stage

### 3.2.1. KNOWLEDGE RETRIEVER

Proteins in downstream tasks often fail outside the PKGs (Zhou et al., 2023), restraining the use of knowledge during fine-tuning. Existing methods thus incorporate knowledge modeling solely during pre-training, leaving the fine-tuning process only guided by task-specific objectives. However, this strategy has several limitations. 1) The optimization objectives of the pre-training and fine-tuning stages are inconsistent (i.e., one is knowledge-guided while the other is knowledge-isolated), causing the pre-training knowledge to be catastrophically forgotten during fine-tuning (Lee et al., 2020). 2) Without PKGs during fine-tuning, these models

fail to explicitly extract helpful knowledge for downstream tasks, leading to unsatisfactory performance. 3) Knowledge graphs are consistently updated (e.g., correcting obsolete knowledge). Existing models cannot adapt to these updates without undergoing retraining. To tackle these challenges, we propose a knowledge retriever that can accurately predict potential knowledge for new proteins, and thus align them with PKGs. This allows the pre-training and fine-tuning stages to directly integrate with knowledge through a unified modeling process, thus unifying the optimization objectives and seamlessly adapting to knowledge updates.

**Generating Candidate Embeddings.** We regard the GO entities in protein knowledge graphs as retrieval candidates. To achieve more accurate and stable retrieval, we integrate the neighboring structure information of each GO entity $g_m$ and generate its candidate embedding as

$$\mathbf{c}_m = \text{MLP}_{aggregation}([\text{MLP}_G(\mathbf{g}_m) : \text{MLP}_G(\mathbf{g}_m^{go}) : \text{MLP}_P(\mathbf{g}_m^{prot})]), \tag{7}$$

where $\mathbf{g}_m$ is the stored embedding of $g_m$. We use $\mathbf{g}_m^{go}$ to incorporate the information of neighboring GO entities of $g_m$, defined as $\mathbf{g}_m^{go} = \frac{1}{|\mathcal{N}_{go}(g_m)|} \sum_{g_k \in \mathcal{N}_{go}(g_m)} \mathbf{g}_k$. Similarly, $\mathbf{g}_m^{prot}$ is used to incorporate the information of $g_m$'s neighboring proteins, defined as $\mathbf{g}_m^{prot} = \frac{1}{|\mathcal{N}_{prot}(g_m)|} \sum_{p_k \in \mathcal{N}_{prot}(g_m)} \mathbf{p}_k$. $\mathcal{N}_{go}(g_m)$ and $\mathcal{N}_{prot}(g_m)$ are respectively the 1-hop neighboring GO entities and 1-hop neighboring proteins of $g_m$. All of $\text{MLP}_{aggregation}$, $\text{MLP}_G$, and $\text{MLP}_P$ are trainable multi-layer perceptrons.

**Retrieval Process.** For each new protein $p_n$, we use a frozen ProtBert to generate its query embedding as $\mathbf{q}_n = \text{MLP}_P(\text{MEAN}(ProtBert(s_n)))$ where $s_n$ is the amino acid sequence of $p_n$. Intuitively, we can traverse the relation set $R$ and the GO entity set $V_{go}$ to find potential knowledge for $p_n$. However, the complexity of this strategy is unacceptable due to the large size of $V_{go}$ (i.e., 47K in ProteinKG25). Fortunately, we observe that each relation only connects with several GO entities in PKGs, inspiring us to reduce the retrieval complexity by finding relation-GO combinations. Specifically, for relation $r_m \in R$, we construct its candidate GO entity set as $\mathcal{E}(r_m) = \{g_m | (p_x, r_m, g_m) \in F\}$. During retrieval, we traverse each $r_m \in R$ and use each of its corresponding candidate GO entity $g_m \in \mathcal{E}(r_m)$ to construct the candidate knowledge $(p_n, r_m, g_m)$. Then we use the TransE objective (Bordes et al., 2013) to score $(p_n, r_m, g_m)$ as

$$\mathbb{S}(p_n, r_m, g_m) = ||\mathbf{q}_n + \tilde{\mathbf{r}}_m - \mathbf{c}_m||_1. \tag{8}$$

$\tilde{\mathbf{r}}_m = \text{MLP}_{rel}(\mathbf{r}_m)$. Finally, we rank all candidate knowledge based on their scores, and then add top-$K$ candidate knowledge into $G$ to align new protein $p_n$ with knowledge graph.

**Training Strategy.** We use triplets $(p_i, r_i, g_i) \in F$ as valid knowledge and by minimizing a margin-based ranking criterion, we hope that valid knowledge can receive lower scores than invalid knowledge. The training objective is defined as

$$\mathcal{L}_{margin} = \text{MAX}(0, \mathbb{S}(p_i, r_i, g_i) - \mathbb{S}(p_i, r_i, g_j) + \gamma). \tag{9}$$

MAX is the maximum operation and $\gamma$ is a hyper-parameter,

controlling the distance between valid and invalid knowledge. $(p_i, r_i, g_j) \notin F$ is invalid knowledge constructed by perturbing $g_i$ in $(p_i, r_i, g_i)$ with a random GO entity $g_j$. Since the retrieval process needs to match information from different modalities (*i.e.,* text descriptions and amino acid sequences), we further propose a cross-modal matching loss to unify the semantic space of embeddings from different modalities.

$$\mathcal{L}_{match} = \text{MAX}(0, ||\text{MLP}_G(\mathbf{g}_i) - \text{MLP}_P(\mathbf{g}_i^{prot})||_1 \\ - ||\text{MLP}_G(\mathbf{g}_i) - \text{MLP}_P(\mathbf{g}_j^{prot})||_1 + \gamma). \tag{10}$$

$\mathbf{g}_j^{prot}$ is the neighboring protein embedding of a randomly sampled GO entity $g_j$. This loss forces the text modality information $\text{MLP}_G(\mathbf{g}_i)$ of $g_i$ to be closer to its corresponding neighboring protein $\text{MLP}_P(\mathbf{g}_i^{prot})$ (*i.e.,* amino acid sequence modality) than other protein information $\text{MLP}_P(\mathbf{g}_j^{prot})$. After jointly optimizing $\mathcal{L}_{margin}$ and $\mathcal{L}_{match}$, the knowledge retriever can accurately predict the potential knowledge for new proteins, enabling its effective alignment with PKGs.

### 3.2.2. TASK-ORIENTED FINE-TUNING

After being aligned with PKGs, new proteins can be encoded with the enhancement of knowledge following Equations (1)-(5). Any related knowledge updates will be perceived when constructing virtual tokens, as they can access the latest PKG to extract knowledge and structures. The downstream task objectives will be used to fine-tune Kara, enabling the protein encoder to extract task-specific knowledge from PKGs via virtual tokens. Note that for each new protein $p_n$, we exclude other new proteins from $\mathcal{N}_1(p_n)$ when constructing structure virtual token $\mathbf{v}_n^p$, to avoid noises.

Moreover, the structure-based regularization can also be seamlessly adapted to the fine-tuning stage. This brings two advantages. 1) Downstream tasks usually lack sufficient training data (Rao et al., 2019). The regularization term can introduce biological function similarities among new proteins as an auxiliary optimization objective, thus effectively avoiding over-fitting. 2) By using this regularization as a unified optimization objective of pre-training and fine-tuning, pre-trained knowledge can avoid being catastrophically forgotten and thus effectively transfer to downstream tasks.

**Complexity.** Compared with vanilla protein language models, the extra complexity of Kara only stems from virtual tokens and retrieval process. Two virtual tokens let the complexity become $O((|S| + 2)^2 d)$ from $O(|S|^2 d)$, where $|S|$ is the length of amino acid sequences. Due to the proposed strategy of finding relation-GO combinations, the complexity of knowledge retriever is $O(|R|k_{max})$, where $|R|$ is the size of the relation set, and $k_{max}$ is the maximum size of the candidate GO entity sets for relations. $k_{max}$ is much smaller than the size of the GO entity set (*e.g.,* In proteinKG25, $k_{max}$ is ~2K, while and the size of GO entity set is 47K).

*Table 1.* Performance in the amino acid contact prediction task. *seq* means the number of amino acids between two selected amino acids. P@L, P@L/2, and P@L/5 denote the precision calculated upon top L (i.e., L most likely contacts), top L/2, and top L/5 predictions, respectively. The best results are **bolded** and The second best results are underlined.

| Models | 6 ≤ seq ≤ 12 | | | 12 ≤ seq ≤ 24 | | | 24 ≤ seq | | |
|---|---|---|---|---|---|---|---|---|---|
| | P@L | P@L/2 | P@L/5 | P@L | P@L/2 | P@L/5 | P@L | P@L/2 | P@L/5 |
| LSTM | 0.26 | 0.36 | 0.49 | 0.20 | 0.26 | 0.34 | 0.20 | 0.23 | 0.27 |
| ResNet | 0.25 | 0.34 | 0.46 | 0.28 | 0.25 | 0.35 | 0.10 | 0.13 | 0.17 |
| Transformer | 0.28 | 0.35 | 0.46 | 0.19 | 0.25 | 0.33 | 0.17 | 0.20 | 0.24 |
| ProtBert | 0.30 | 0.40 | 0.52 | 0.27 | 0.35 | 0.47 | 0.20 | 0.26 | 0.34 |
| ESM-1b | 0.38 | 0.48 | 0.62 | 0.33 | 0.43 | 0.56 | 0.26 | 0.34 | 0.45 |
| ESM-2 | 0.40 | 0.50 | 0.62 | 0.35 | 0.44 | 0.56 | 0.27 | 0.35 | 0.45 |
| OntoProtein | 0.37 | 0.46 | 0.57 | 0.32 | 0.40 | 0.50 | 0.24 | 0.31 | 0.39 |
| KeAP | 0.41 | 0.51 | 0.63 | 0.36 | 0.45 | 0.54 | 0.28 | 0.35 | 0.43 |
| Kara | **0.45** | **0.55** | **0.65** | **0.39** | **0.48** | **0.59** | **0.31** | **0.39** | **0.48** |

## 4. Experiments and Analyses

We evaluate Kara across six downstream tasks, such as amino acid contact prediction, homology detection, and stability prediction. Our analysis includes hyper-parameter sensitivity, component-wise ablations, detailed examinations of the generalization ability to unseen knowledge, and the analysis of model robustness to PKG incompleteness. Detailed task descriptions are in Appendix D. Experimental settings and implementation details are in Appendix E. Results are averaged over 3 independent runs.

### 4.1. Amino Acid Contact Prediction

**Overview.** This task aims to predict whether two amino acids within a protein are in contact, which is a token-level classification task (Rao et al., 2019). Following Zhou et al. (2023), we use variants of LSTM, ResNet, and Transformer proposed by the TAPE benchmark (Rao et al., 2019), pre-trained language models ProtBert (Ahmed et al., 2022), ESM-1b (Rives et al., 2021), and knowledge-enhanced model OntoProtein (Zhang et al., 2022a) as baselines. The state-of-the-art knowledge-enhanced model KeAP (Zhou et al., 2023) and the recent powerful protein language model ESM-2-30t (Lin et al., 2023) are also used for comparison.

**Results.** As shown in Table 1, Kara outperforms baselines by large margins in short- ($6 \leq seq \leq 12$), medium- ($12 \leq seq \leq 24$), and long-range ($24 \leq seq$) contact predictions, achieving on average 9.5% and 11.0% improvements in P@L and P@L/2 metrics. Compared with the state-of-the-art PKG-enhanced model KeAP, Kara consistently surpasses it, especially in challenging long-range predictions. This is due to Kara's contextualized virtual tokens, allowing each amino acid token to explicitly extract task-oriented knowledge from PKG. However, KeAP fails to incorporate knowledge during the fine-tuning stage.

### 4.2. Protein-Protein Interaction Identification

**Overview.** Protein-protein interaction (PPI) identification aims to predict the interaction types of protein pairs and is sequence-level classification. Experiments are done on three

*Table 2.* Performance in the protein-protein interaction identification task. BFS (breadth-first search (BFS) and DFS (depth-first search) indicate the strategies to generate. We use F1 score as the metric.

| | SHS27K | | | SHS148K | | | STRING | | |
|---|---|---|---|---|---|---|---|---|---|
| **Models** | BFS | DFS | Avg | BFS | DFS | Avg | BFS | DFS | Avg |
| DNN-PPI | 48.09 | 54.34 | 51.22 | 57.40 | 58.42 | 57.91 | 53.05 | 64.94 | 59.00 |
| DPPI | 41.43 | 46.12 | 43.77 | 52.12 | 52.03 | 52.08 | 56.68 | 66.82 | 61.75 |
| PIPR | 44.48 | 57.80 | 51.14 | 61.83 | 63.98 | 62.91 | 55.65 | 67.45 | 61.55 |
| GNN-PPI | 63.81 | 74.72 | 69.27 | 71.37 | 82.67 | 77.02 | 78.37 | 91.07 | 84.72 |
| ProtBert | 70.94 | 73.36 | 72.15 | 70.32 | 78.86 | 74.59 | 67.61 | 87.44 | 77.53 |
| ESM-1b | 74.92 | 78.83 | 76.88 | 77.49 | 82.13 | 79.31 | 78.54 | 88.59 | 83.57 |
| ESM-2 | 75.05 | 79.55 | 77.30 | 77.19 | 83.34 | 80.26 | 81.32 | 89.19 | 85.30 |
| OntoProtein | 72.26 | **78.89** | 75.58 | 75.23 | 77.52 | 76.38 | 76.71 | **91.45** | 84.08 |
| KeAP | 78.58 | 77.54 | 78.06 | 77.22 | 84.74 | 80.98 | 81.44 | 89.77 | 85.61 |
| Kara | **81.18** | 78.85 | **80.01** | **79.62** | **86.02** | **82.82** | **82.73** | 92.46 | **87.59** |

*Table 3.* Protein homology detection and stability prediction.

| Models | Homology | Stability |
|---|---|---|
| LSTM | 0.26 | 0.69 |
| ResNet | 0.17 | 0.73 |
| Transformer | 0.21 | 0.73 |
| ProtBert | 0.29 | 0.78 |
| ESM-1b | 0.11 | 0.77 |
| ESM-2 | 0.13 | 0.80 |
| OntoProtein | 0.24 | 0.75 |
| KeAP | 0.29 | 0.80 |
| Kara | **0.32** | **0.83** |

*Table 4.* Result of protein-protein binding affinity prediction.

| Models | Affinity ↓ |
|---|---|
| PIPR | 0.63 |
| ProtBert | 0.58 |
| ESM-1b | 0.50 |
| ESM-2 | 0.50 |
| OntoProtein | 0.59 |
| KeAP | 0.52 |
| Kara | **0.50** |

widely-used datasets SHS27K (Chen et al., 2019), SHS148K (Chen et al., 2019), and STRING (Lv et al., 2021). 7 types of interactions are included. As in Zhang et al. (2022a), we use DPPI (Hashemifar et al., 2018), DNNPPI (Li et al., 2018), PIPR (Chen et al., 2019), GNN-PPI (Lv et al., 2021) as baselines. The LM baselines are ProtBert, ESM-1b, ESM-2. Knowledge-enhanced baselines are KeAP and OntoProtein.

**Results.** From Table 2, we see that Kara outperforms baselines on nearly all datasets, showing its effectiveness in accurately understanding the relationships between proteins. An interesting observation is that the performance gains of KeAP compared with OntoProtein are very small on STRING dataset. As suggested in Zhou et al. (2023), this is because the large number of fine-tuning data in the STRING dataset reduces the impact of knowledge modeling in pre-training. In contrast, Kara incorporates knowledge modeling in both pre-training and fine-tuning stages, thus avoiding catastrophically forgetting pre-trained knowledge.

### 4.3. Homology Detection and Stability Prediction

**Overview.** Homology detection aims to predict the remote homology of protein, which is a sequence-level classification task. We follow the datasets and experimental settings of Hou et al. (2018), and ask the model to predict the right fold type of protein from 1,195 different types. We report average accuracy on the fold-level heldout set. Stability prediction aims to predict the intrinsic stability of a protein, which is a sequence-level regression task. As in Rocklin et al. (2017), we use Spearman's rank correlation scores for evaluation. The same baselines are used as in Table 1.

**Results.** As illustrated in Table 3, existing knowledge-enhanced models (*i.e.,* OntoProtein and KeAP) cannot outperform traditional language models. Previous works (Zhang et al., 2022a) attributed this failure to the lack of sequence-level objectives during pre-training. Instead, using structure-based regularization, Kara incorporates knowledge-based relevance (*i.e.,* function similarity) among proteins as a unified sequence-level objective in both pre-training and fine-tuning stages, thus achieving better results.

### 4.4. Protein-Protein Binding Affinity Prediction

**Overview.** This task aims to map each pair of proteins to a real value to denote their binding affinity changes, *i.e.,* a sequence-level regression. As in Unsal et al. (2022), we use Bayesian ridge regression to the element-wise multiplication of protein embeddings for prediction. The SKEMPI dataset (Moal & Fernández-Recio, 2012) is used. Result is reported based on mean square error of 10-fold cross-validation. We use the same baselines as (Zhou et al., 2023), additionally with KeAP and ESM-2.

**Results.** As shown in Table 4, both knowledge-enhanced models fail to outperform ESM-1b. This is because protein structure plays a vital role in this task (Unsal et al., 2022), and existing models overlook the modeling of protein structures, while ESM-1b achieves it via its network architecture. Kara achieves competitive performance with ESM-1b, since the protein knowledge graph contains the description of the protein structure, and Kara can directly inject such knowledge into protein embeddings via the virtual tokens.

### 4.5. Semantic Similarity Inference

**Overview.** This task evaluates models' ability to extract biomolecular functional similarity among proteins. As in Unsal et al. (2022), we use biological process (BP) and cellular component (CC) to divide protein attributes into two groups and calculate Lin similarity in each group as the ground-truth similarity. We then calculate Manhattan similarity between protein embeddings for prediction. The Spearman's rank correlation between these similarities is calculated as the metric. We include another powerful protein language model MSA Transformer (Rao et al., 2021) as baseline.

**Results.** Table 5 shows that Kara outperforms knowledge-enhanced models on both BP and CC. This can be attributed to the explicit incorporation of GO entity information in Kara, which describes the functionality of proteins. Kara is unable to outperform ESM-1b on BP, potentially because of the larger number of parameters of ESM-1b. However, it still outperforms the larger model ESM-1b on CC, indicating its effectiveness in explicitly incorporating GO entities.

Table 5. Performance in the semantic similarity inference task.

| Models | BP | CC |
|---|---|---|
| MSA Transformer | 0.31 | 0.30 |
| ProtBert | 0.35 | 0.36 |
| ESM-1b | **0.42** | 0.37 |
| ESM-2 | 0.41 | 0.39 |
| OntoProtein | 0.36 | 0.36 |
| KeAP | 0.41 | 0.40 |
| Kara | 0.41 | **0.41** |

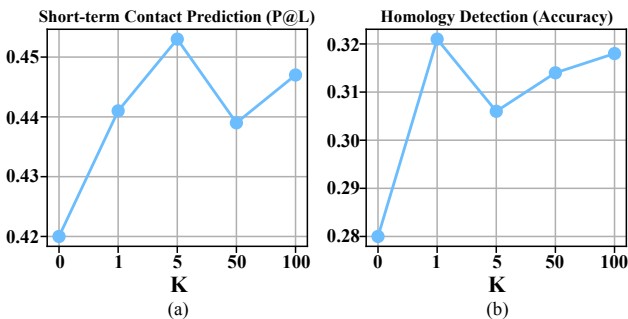

Figure 3. Kara with different numbers of knowledge $K$.

### 4.6. Analysis of Kara

**Ablation and Variants.** In Table 6 we study the effectiveness of each component. We see that virtual tokens, structure-based regularization, and knowledge retriever are essential to achieve good performance. Specifically, removing contextualized virtual tokens makes Kara unable to incorporate knowledge explicitly, thus degrading its performance in protein-protein binding affinity prediction task, which requires the property understanding of proteins. After removing structure-based regularization, Kara fails to integrate function similarities into sequence-level protein embeddings, resulting in performance degradation in sequence-level tasks, *e.g.,* homology detection and stability prediction.

To assess the effectiveness of our proposed knowledge retriever, we compare it to a variant that uses a protein similarity-based retriever. In this variant, we use the frozen ProtBert model to calculate embedding similarities between new proteins and those in the PKG, selecting the top-$K$ similar proteins and using their embeddings as virtual tokens. However, this approach does not outperform Kara. The reason is that similarity-based retrievers struggle to accurately predict associated knowledge (*i.e.,* gene descriptions) for proteins, but proteins with similar sequences can have different functions, so this approach may introduce irrelevant protein information as noise during encoding.

**Hyper-parameter Analysis.** During pre-training, we use the ground-truth knowledge graph structure to construct the virtual tokens. However, in the fine-tuning stage, because the new proteins are not included in the protein knowledge graph, we need to use the knowledge retriever to predict its top-$K$ potential knowledge to construct the virtual tokens for fine-tuning and inference, where $K$ is a hyper-parameter used to control the amount of predicted potential knowledge incorporated. Because the predicted potential knowledge can bring additional information but also inevitable noise, in this part we study how $K$ affects the performance of Kara. As shown in Figure 3, the performance improves across different tasks when $K$ increases from 0 to 1, showcasing the value of incorporating knowledge into protein representations. As $K$ continues to increase, performance fluctuates due to the introduction of noise from additional knowledge. Nevertheless, it still outperforms the variant without

knowledge (*i.e.,* $K$=0), demonstrating Kara's ability to effectively extract useful knowledge for downstream tasks.

**Backbone Model Analysis.** We follow Zhou et al. (2023) and use ProtBert as the backbone model for a fair comparison to baselines. Here we further adapt different backbones to present the flexibility of Kara in Table 7. Results show that Kara can flexibly adapt different backbones and produce competitive performance in different tasks. Specifically, by using larger protein models as backbone, Kara can further improve the results, demonstrating the adaptability of Kara.

**Protein Knowledge Graph Analysis.** We follow Zhang et al. (2022a); Zhou et al. (2023) and use commonly adopted ProteinKG25 as PKG for a fair comparison to baselines. Here we analyze the effect of incomplete PKG on Kara's performance in Table 8. Results show that Kara with incomplete PKG still outperforms OntoProtein and KeAP with full PKG, showing Kara's robustness. We attribute the outperformance to knowledge retriever and virtual tokens, which well integrate knowledge updates into model learning.

### 4.7. Analysis of Knowledge Retriever

**Ablation Study.** The accurate knowledge retriever is extremely important for Kara's performance in downstream tasks. Here we analyze how different components and hyper-parameters affect the retrieval performance of knowledge retriever. Knowledge retriever is trained on ProteinKG25 knowledge graph, and we use the randomly sampled 2,000 proteins as test set to select the best model. During evaluation, for each test protein $p_t$ we first traverse each relation $r \in R$ to construct query pairs $(p_t, r, ?)$, and then use the knowledge retriever model to score the corresponding candidate knowledge $(p_t, r, g_i^r)$, where $g_i^r$ is the candidate GO entity from $\mathcal{E}(r)$. After traversing all the relations, we rank candidate knowledge based on their scores and calculate Precision@n (P@n) to evaluate the retrieval performance, indicating how much knowledge on the top-n ranked candidates is valid (*i.e.,* exists in the protein knowledge graph).

**Hyper-parameter Analysis.** In Table 9, "without structure information" means that we remove neighbor information

*Table 6.* Ablation study and performance of variants.

| Tasks | Concate ($6 \leq seq \leq 12$) | PPI (STRING) | Homology | Stability | Affinity ↓ |
|---|---|---|---|---|---|
| w/o contextualized virtual tokens | 0.42 | 85.16 | 0.28 | 0.81 | 0.55 |
| w/o structure-based regularizations | 0.43 | 86.49 | 0.30 | 0.80 | 0.52 |
| Retrieval based on the protein sequence similarities | 0.43 | 85.33 | 0.29 | 0.79 | 0.57 |
| Kara | **0.45** | **87.59** | **0.32** | **0.83** | **0.50** |

*Table 7.* Performance with different backbone models.

| Models | Contact ↑ ($6 \leq seq \leq 12$) | Homology ↑ | Stability ↑ | Affinity ↓ |
|---|---|---|---|---|
| OntoProtein | 0.460 | 0.240 | 0.750 | 0.590 |
| KeAP | 0.510 | 0.290 | 0.800 | 0.520 |
| Kara (ProtBert) | 0.553 | 0.323 | 0.830 | **0.501** |
| Kara (ProteinBert) | 0.556 | 0.318 | 0.824 | 0.506 |
| Kara (ESM-1b) | **0.563** | **0.327** | **0.833** | 0.510 |

*Table 10.* Performance comparison of different models after pre-training and after task fine-tuning.

| Models | After Pre-training | | After Task Fine-tuning | |
|---|---|---|---|---|
| | Precision | Similarity | Precision | Similarity |
| OntoProtein | 0.712 | 0.901 | 0.621 | 0.632 |
| KeAP | 0.705 | 0.918 | 0.645 | 0.677 |
| w/o structure-based regularization | 0.722 | 0.906 | 0.624 | 0.749 |
| w/o contextualized virtual tokens | 0.713 | 0.902 | 0.676 | 0.816 |
| Kara | 0.738 | 0.934 | 0.725 | 0.968 |

*Table 8.* Performance with incomplete protein knowledge graph.

| Models | Contact ↑ ($6 \leq seq \leq 12$) | Homology ↑ | Stability ↑ | Affinity ↓ |
|---|---|---|---|---|
| OntoProtein (full KG) | 0.460 | 0.240 | 0.750 | 0.590 |
| KeAP (full KG) | 0.510 | 0.290 | 0.800 | 0.520 |
| Kara (50% KG) | 0.540 | 0.316 | 0.823 | 0.511 |
| Kara (70% KG) | **0.546** | **0.322** | **0.828** | **0.503** |

but also help filter out irrelevant GO candidates, thus improving retrieval accuracy. As shown in Figure 4, the higher neighbor sampling number helps achieve better retrieval result. We further analysis the generalization ability of our knowledge retriever to unseen knowledge in Appendix F.

### 4.8. Case Study

**Mitigating Catastrophic Forgetting.** To evaluate Kara's effectiveness in mitigating catastrophic forgetting, we designed two experiments. The first measures the similarity between the embeddings of two proteins with the same attribute knowledge—a higher cosine similarity indicates better retention of knowledge information. The second requires the model to identify, from a set of candidate proteins, the one sharing attribute knowledge with a given protein. Higher accuracy suggests better embedding and preservation of knowledge information.

As shown in Table 10, OntoProtein, KeAP, and Kara all perform well after pre-training, confirming their ability to learn attribute knowledge. Kara achieves the highest performance, demonstrating its superior knowledge acquisition capability. After fine-tuning on downstream tasks, Kara's performance remains stable, whereas OntoProtein and KeAP show significant drops, indicating that they lose some of the knowledge acquired during pre-training. Furthermore, removing the structure loss or virtual token leads to performance degradation after fine-tuning, highlighting the importance of unified knowledge integration in mitigating catastrophic forgetting.

*Table 9.* Ablation study results of the knowledge retriever.

| Metrics | P@1 | P@5 |
|---|---|---|
| Without structure information | 0.681 | 0.669 |
| Without cross-modal matching | 0.733 | 0.721 |
| Without relation-GO combinations | 0.649 | 0.538 |
| Original | **0.821** | **0.795** |

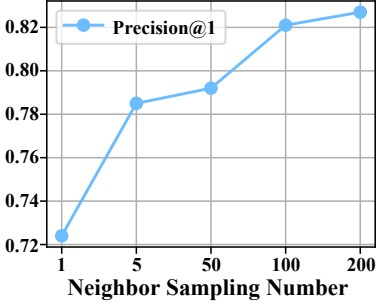

*Figure 4.* Performance of knowledge retriever with different neighbor sampling numbers.

in candidate GO embeddings (Equation (7)), and "without cross-modal matching" means that the knowledge retriever is only optimized based on $\mathcal{L}_{margin}$. Both components are beneficial to the retrieval performance. "Without relation-GO combinations" means that for each relation, we use the whole GO entity set as candidates during retrieval. The worse performance of this variant shows that relation-GO combination strategy can not only reduce the retrieval time consumption,

**Error Analysis.** As shown in the first and second lines of Figure 5, Kara outperforms KeAP in predicting contacts for proteins with short sequences. However, as the sequence length increases, both Kara and KeAP struggle to accurately align with the ground-truth contact map. This limitation may stem from the lack of protein structural information modeling, which is crucial for handling long-sequence proteins.

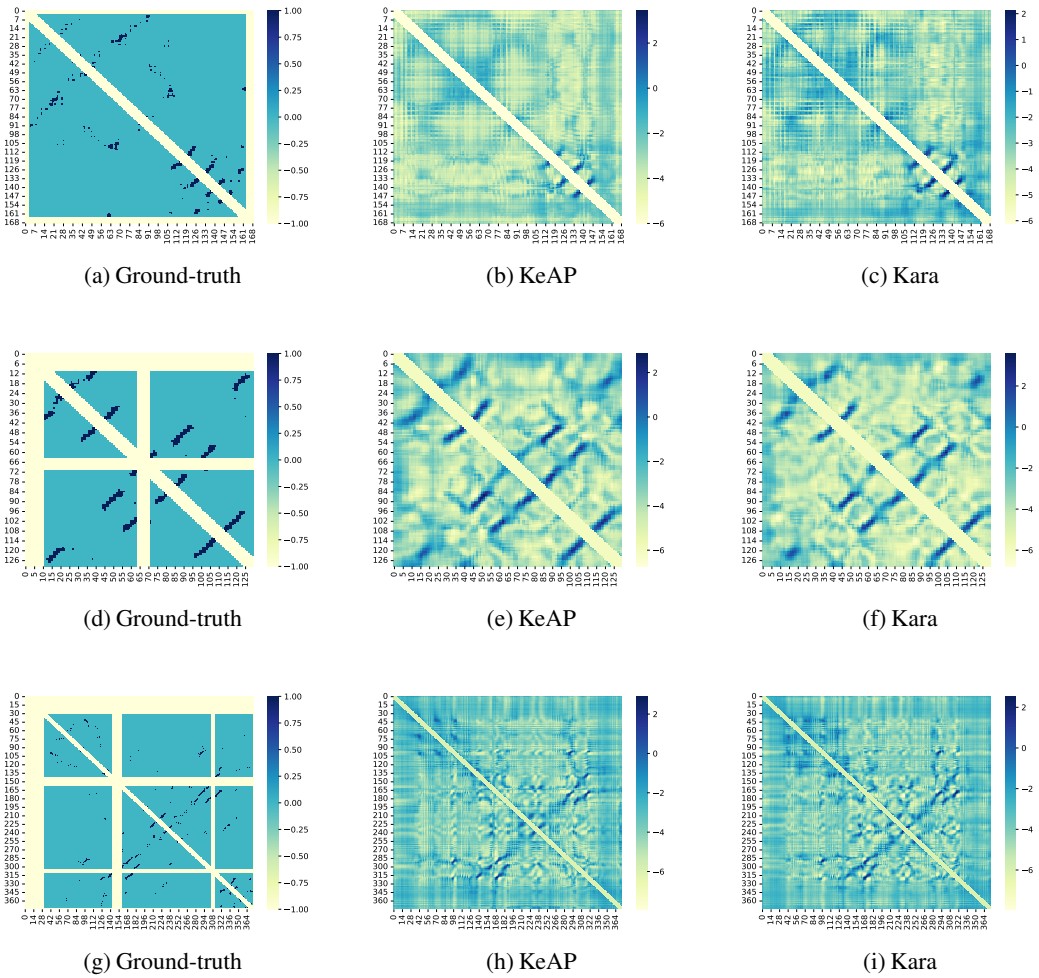

*Figure 5.* Ground-truths and predicted probability contact maps.

## 5. Related Work

Protein representation learning has attracted much attention due to the rapid development of language models. Existing works treat amino acid sequences as token sequences, and train the language model with either supervision (Bepler & Berger, 2019) or self-supervised objective (Alley et al., 2019; Rao et al., 2019; Xiao et al., 2021; Ahmed et al., 2022; Unsal et al., 2022; Lin et al., 2023; Brandes et al., 2022). However, they ignore factual knowledge (*e.g.,* gene descriptions of proteins), resulting in inferior representations. Recently, OntoProtein (Zhang et al., 2022a) incorporates PKG by proposing a hybrid encoder. KeAP (Zhou et al., 2023) extends it by performing token-level knowledge exploration via cross-attention module. However, both are limited by ignoring graph structure and task-oriented knowledge modeling. Very recently, GOProteinGNN (Kalifa et al., 2024) explores the benefit of graph structure. However, it still suffers from inconsistent objectives and fails to consider the high-order

relationships. Instead, Kara explicitly injects high-order knowledge during both pre-training and fine-tuning stages.

Some models incorporate other modalities to improve protein representations (Chen et al., 2023a). For example, Otter-Knowledge (Lam et al., 2023) designs knowledge graphs for broadly biomedical concepts. ProtST (Xu et al., 2023) infers protein representations from biomedical texts, but with no graph structure. Kara captures text descriptions together with knowledge graphs for high-order knowledge incorporation.

## 6. Conclusion and Future Work

We develop a retrieval-augmented language model for knowledge-aware protein representation learning, which achieves direct integration of high-order knowledge graphs and protein language models. Experiments show Kara's superiority in 6 downstream tasks. A future direction is to integrate 3D structure for protein representation learning.

## Acknowledgements

This work is supported by the National Natural Science Foundation of China (No. 62276047) and Guangxi Key Research and Development Program (No. Guike AB24010112).

## Impact Statement

This paper presents work whose goal is to advance the field of AI for protein science. Protein representation learning aims to infer low-dimensional representations using amino acid sequences of proteins as well as their multi-modal data, such as protein knowledge graph and textual descriptions. The learned protein representations could fulfill diverse downstream tasks. One societal consequence of our work is to infer protein properties and functions, *e.g.,* stability, for a previously unseen protein, which facilitates scientific discovery process. We do not foresee any undesired implications stemming from our work.

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

*Table 11.* Important notations and descriptions.

| Notation | Description |
|---|---|
| $G$ | A protein knowledge graph |
| $V_p, V_{go}, R$ | Protein set, GO entity set, and relation set in $G$ |
| $F$ | Set of triplets (*i.e.,* knowledge) in $G$ |
| $p_i, r_j, g_k$ | A protein, a GO entity, and a relation |
| $s_i, s_i^m$ | The amino acid sequence of protein $p_i$, and each amino acid in $s_i$ |
| $\mathbf{p}_i, \mathbf{r}_j, \mathbf{g}_k$ | Stored pre-trained embeddings of protein $p_i$, relation $r_j$, and GO entity $g_k$ (see Section 2) |
| $\mathbf{v}_i^k, \mathbf{v}_i^p$ | Knowledge virtual token and structure virtual token of protein $p_i$ |
| $\mathbf{S}_i, \mathbf{S}_i^L$ | Input embedding sequence of protein $p_i$, embedding sequence at the $L$-th layer |
| $\tilde{\mathbf{p}}_i$ | Encoded embedding of protein $p_i$ by Kara |
| $\mathbf{g}_m^{go}, \mathbf{g}_m^{prot}$ | Neighboring GO entity embedding and neighboring protein embedding of GO entity $g_m$ |
| $\mathbf{q}_n$ | Query embedding corresponds to new protein $p_n$ |
| $\tilde{\mathbf{r}}_m$ | Query embedding corresponds to relation $r_m$ |
| $\mathbf{c}_m$ | Candidate embedding corresponds to GO entity $g_m$ |
| $\mathbb{S}(\cdot)$ | Score function |
| $\mathrm{MLP}(\cdot)$ | Trainable multi-layer perceptron |
| $N_1(p_i)$ | One-hop GO entities with relations of protein $p_i$ |
| $N_2(p_i)$ | Two-hop connected proteins of protein $p_i$ |
| $N_{go}(g_m)$ | One-hop neighboring GO entities of GO entity $g_m$ |
| $N_{prot}(g_m)$ | One-hop neighboring proteins of GO entity $g_m$ |
| $\mathcal{E}(r_m)$ | Candidate GO entity set corresponding to relation $r_m$ |

## A. Mathematical Notations

Here we summarize main mathematical notations used in the paper in Table 11.

## B. Dataset Description

We train the proposed Kara using the ProteinKG25 knowledge graph (Zhang et al., 2022a), consistent with previous knowledge-enhanced models to achieve a fair comparison. ProteinKG25 includes about 4.5 million triplets describing relationships between protein and gene ontology (GO) entities, and 100K triplets describing relationships between GO entities. There are 31 kinds of relations, 600K proteins, and 50K GO entities in ProteinKG25. Each GO entity in ProteinKG25 can be a molecule, a cellular component, or a biological process, and each protein in ProteinKG25 has an average of 8.64 relations. Following the strategy provided by Zhou et al. (2023), we removed proteins appearing in the datasets of downstream tasks to avoid data leakage. The raw data of ProteinKG25 can be found in `https://www.zjukg.org/project/ProteinKG25/`.

## C. Detailed Analysis of Kara

**Differences Compared with Retrieval-augmented LMs in Other Fields.**

- First, some previous works in the question-answering field also use virtual tokens to incorporate knowledge information (Zhang et al., 2022b; Sun et al., 2021; Zhang et al., 2024). They typically assume every encoding objective has existed in KG and the knowledge information can be directly extracted after matching corresponding entities. However, in the protein-encoding scenario, many under-studied proteins do not exist in KG, making the previous "matching and extracting" strategy not work. To tackle this challenge, we propose a novel knowledge retriever to predict potential gene descriptions for new proteins, which enables our model to generalize to unseen encoding objectives (where previous work fell short).
- Second, some recent methods also propose using a retriever to find related entities from KG to enhance LLM generation (He et al., 2024; Hu et al., 2025). However, they are designed for general KGs and cannot handle unique challenges for protein knowledge graphs. Specifically, protein KGs contain two types of entities with different modality information, requiring the retrieval process to consider multi-modal information alignment. Additionally, it contains a large amount of different textual gene descriptions, bringing a large candidate space with complex semantics. Our knowledge retriever is specially designed to solve these challenges with multi-modal matching loss and relation-go combination strategies.
- Third, some previous works also integrate knowledge and structural information within KGs into training objectives (Wang et al., 2021; Li et al., 2022; Pan et al., 2024). They are typically designed for document encoding where the entities in KG are words that appeared in documents. They use structural information to assign mask possibilities for different

words during masked language modeling or train the model to predict graph neighbors. However, in the protein-encoding scenario, both the encoding objective and entities in KG are protein sequences, making previous training strategies not work. Moreover, they only predict one-hop neighbors, which overlooked high-order structural relevance in their objective functions. However, high-order relevance is important for protein encoding since it indicates the functional similarity between proteins.

In summary, Kara contains several unique technical designs (*e.g.,* knowledge retriever and structure-based regularizations) to solve special challenges in protein-encoding scenarios, making it different from previous retrieval-augmented LMs and being an enhanced paradigm for integrating protein knowledge graphs into protein language models.

**Detailed Comparison to KeAP.**

- From how to integrate knowledge into language models. KeAP implicitly embeds knowledge within the parameters of the language model. Specifically, during pre-training, it uses the protein language model to encode a protein's amino acid sequence into an embedding. A Transformer-based decoder then takes this embedding along with related knowledge to predict masked amino acid tokens. KeAP proposes that this knowledge-guided pre-training approach helps retain knowledge within the model parameters. However, language models often struggle to retain knowledge precisely. Additionally, KeAP processes each piece of knowledge independently, failing to integrate the complete knowledge context of proteins. Kara directly uses knowledge of each protein as a part of the language model's input. As described in Section 3, Kara summarizes 1-hop neighbors of a protein (gene descriptions) as "knowledge virtual tokens" and 2-hop neighbors (functionally similar proteins) as "structure virtual tokens". These virtual tokens are then concatenated with the amino acid sequence to form the model input. This approach not only can input precise knowledge information to the language model, but also provides a broader knowledge context by leveraging neighboring information.
- From how to pre-train the language model. KeAP employs a decoder to predict masked amino acid tokens using knowledge input and embeddings encoded by protein language model. However, the Transformer-based decoder introduces significant training complexity and a large number of parameters. Additionally, KeAP's pre-training overlooks the protein relevance provided by the KG structure, leading to insufficient knowledge exploitation. Kara predicts masked amino acid tokens directly using the protein language model with the prompt of virtual tokens. This eliminates the need for a decoder, reducing both training complexity and parameter size. Furthermore, Kara is also trained to embed the functionally similar proteins closer together in embedding space, integrating high-order graph structural relevance (*i.e.,* functional similarity) into protein representations.
- From how to encode new proteins. Since KeAP assumes that knowledge has been embedded within parameters of the language model, they directly input the amino acid sequence of new protein into the pre-trained language model to get its embedding, which suffers from imprecise knowledge information, and fails to adapt to knowledge updates. Kara proposes a novel knowledge retriever, retrieving related knowledge for each new protein and summarizing the retrieved knowledge as virtual tokens to input into the language model, which can integrate precise knowledge into the protein language model. Moreover, any updates of the related knowledge of a protein can be perceived by the knowledge retriever during retrieving, and then integrated during encoding via the virtual tokens, ensuring that Kara can always use the most current knowledge for encoding.
- From model complexity. Due to the incorporation of a Transformer-based decoder, the additional time complexity of KeAP compared with vanilla protein language models is $O(|S|^2 \times d)$, where $|S|$ is the length of protein amino acid sequence (typically $\geq 500$), and $d$ is the embedding hidden size (usually 768 or 1024). Kara's additional time complexity, compared with vanilla protein language models, arises only from the virtual tokens (increasing from $O(|S|^2 \times d)$ to $O((|S|+2)^2 \times d)$) and the retrieval process ( $O(|R \times k|)$ ), where $|R \times k|$ is much smaller than $|S|^2 \times d$. Therefore, the time complexity of Kara is much smaller than that of KeAP.
- From parameter number. For KeAP, the incorporation of a Transformer-based decoder brings a large number of parameters, including $Q$, $K$, $V$, and $O$ weight matrices for $n$ heads, the MLP for the multi-head mechanism, layer normalization, etc. The additional parameter of Kara only comes from four projection matrices: $\text{MLP}_{knowledge}$, $\text{MLP}_{struture}$, $\text{MLP}_G$, and $\text{MLP}_P$, which is much smaller than KeAP.

# D. Downstream Task Definitions

**Amino Acid Contact Prediction.** This is a pairwise token-level matching task, where each pair of input amino acids $(s^i, s^j)$ from a protein sequence $s$ is mapped to a label $y_{i,j} \in \{0, 1\}$, indicating whether they are in contact or not ($< 8$Å apart). Accurate contact maps can facilitate robust modeling of full 3D protein structure (Kim et al., 2014). Following previous

works (Zhou et al., 2023), we use data that comes from ProteinNet (AlQuraishi, 2019) and report precision on the ProteinNet CASP12 test set, which is a standard metric reported in CASP (Moult et al., 2018).

**Protein-protein Interaction Identification.** This is a pairwise sequence-level classification task. Given a pair of proteins $(p_i, p_j)$, the model aims to predict the interaction types $y_{i,j}$ between them. Similar to previous works (Zhou et al., 2023), 7 types of interactions are included in our experiments, which are reaction, binding, post-translational modifications, activation, inhibition, catalysis, and expression. Each protein pair may belong to several types simultaneously so this is a multi-label classification problem. We use three widely-used datasets SHS27K (Chen et al., 2019), SHS148K (Chen et al., 2019), and STRING (Lv et al., 2021) in our experiments, where SHS27K and SHS148K can be regarded as two subsets of STRING, which remove proteins with no more than 50 amino acids or $\geq 40\%$ sequence identity. The F1 score is used as the evaluation metric for this task.

**Homology Detection.** This is a sequence-level classification task where each input protein $p$ is mapped to a label $y \in \{1, 2, ..., 1195\}$ based on its representation generated by protein language models, which indicates its possible protein fold. This task requires the evolutionary understanding of proteins and thus is valuable in microbiology and medicine (*e.g.,* discover new CAS enzymes (Liu et al., 2019)). We follow the previous works and use data from Hou et al. (2018). By holding out entire evolutionary groups from the training set, the model is required to generalize across evolutionary gaps. Same as Hou et al. (2018), we report accuracy on the fold-level heldout set.

**Stability Prediction.** This is a sequence-level regression task. Each input protein $p$ is mapped as a number $y \in \mathbb{R}$, which represents the most extreme conditions under which the protein maintains its structure above a concentration threshold, serving as a proxy for its intrinsic stability. Measuring the stability of proteins is important for finding top candidates from expensive protein engineering experiments (Rao et al., 2019). We use the data provided by Rocklin et al. (2017), where the training set includes proteins from four rounds of experimental design, while the test set contains proteins that are Hamming distance-1 neighbors of the top candidates. We report the Spearman's rank correlation scores on the test set to evaluate the model performance.

**Protein-protein Binding Affinity Prediction.** This is a pairwise sequence-level regression task that maps each pair of proteins $(p_i, p_j)$ as a real value $y \in \mathbb{R}$, indicating the binding affinity changes between them. This task evaluates how well a protein representation can predict changes in binding affinity resulting from protein mutations, thus being valuable for many downstream applications such as drug design (Reidenbach, 2024). Following Unsal et al. (2022), we use Bayesian ridge regression to the element-wise multiplication of protein embeddings for predicting the binding affinity. The SKEMPI dataset (Moal & Fernández-Recio, 2012) is used and the performance is reported based on the mean square error of 10-fold cross-validation.

**Semantic Similarity Inference.** This is a pairwise sequence-level regression task, which evaluates how well protein language models can capture information about biomolecular functional similarity between proteins. In this task, we emphasize the biological process (BP) and cellular component (CC) categories similar to previous works (Unsal et al., 2022). We first use BP and CC to divide protein attributes into two groups and calculate the Lin similarity in each group as the ground-truth similarity. We then calculate the Manhattan similarity between protein embeddings as the prediction. The Spearman's rank correlation between these similarities is calculated as the metric.

# E. Experimental Details

**Experimental Settings.** Same as previous knowledge-enhanced protein language models such as KeAP and OntoProtein, we use the ProtBert model[1] as the backbone of the protein encoder within Kara for a fair comparison. The text descriptions of GO entities and relations are encoded by the PubMedBert model[2], which is also consistent with previous works. While generating the pre-trained embeddings of items in the protein knowledge graph (see Section 2), we represent each item as averaging the embeddings of its amino acid or word tokens. Our model is implemented with Python and we refer to the official code released by Zhou et al. (2023) to implement the downstream task experiments. All tasks use standard datasets and metrics, consistent with previous works, to ensure a fair comparison. Note that since the train/valid/test set splittings of SHS27K, SHS148K, and STRING datasets are not provided, we use the official code released by Lv et al. (2021) to split each dataset with three different random seeds, and the average performance of each dataset is reported. All the experiments are conducted on NVIDIA A40 with 48 GB memory.

**Pre-training Implementation Details.** In the pre-training stage, we set the protein encoder within Kara (*i.e.,* a PortBert model) as full-parameter trainable similar to previous works (Zhang et al., 2022a). We only use proteins and knowledge preserved

---

[1] https://huggingface.co/Rostlab/prot_bert
[2] https://huggingface.co/microsoft/BiomedNLP-BiomedBERT-base-uncased-abstract-fulltext

*Table 12.* Hyper-parameter settings for different downstream tasks.

| Tasks | Train Steps | Batch Size | $K$ | $\mathcal{L}_{reg}$ | Learning Rate | Gradient Accumulation Step |
|---|---|---|---|---|---|---|
| Contact | 30,000 | 1 | 5 | False | 3e-5 | 8 |
| Homology | 2,200 | 2 | 1 | True | 4e-5 | 16 |
| Stability | 4,800 | 5 | 5 | True | 1e-5 | 16 |

*Table 13.* Performance of the knowledge retriever on unseen knowledge.

| Models | Hits@1 | Hits@3 | Hits@10 |
|---|---|---|---|
| without PubMedBert fine-tuning | 0.430 | 0.608 | 0.796 |
| with PubMedBert fine-tuning | 0.495 | 0.683 | 0.859 |

in the ProteinKG25 knowledge graph to pre-train Kara, where the maximum token length is set as 1024 for proteins and 512 for text descriptions. For each protein, we randomly select 10 knowledge and 10 high-order connected proteins respectively from $\mathcal{N}_1$ and $\mathcal{N}_2$ to construct its virtual tokens. The margin $\gamma$ is set as 5 and the number of negative samples is set as 2. We set the batch size to 4 with the maximum number of update steps to 10,000, and the gradient accumulation step to 16. The learning rate is set as 1e-6 and we use AdamW (Loshchilov & Hutter, 2019) for optimization. The weight decay is set as 1e-2.

**Knowledge Retriever Implementation Details.** In the knowledge retriever, we set the sampling number of neighbors during the candidate embedding generation as 100. Similar to the pre-training stage, the maximum token length is 1024 for proteins and 512 for text descriptions. To train the knowledge retriever, we randomly sample 2,000 proteins as well as their associated knowledge from the ProteinKG25 knowledge graph as the test set, and the remaining proteins are used as training data. The best knowledge retriever model is selected based on the Precision@5 metric on the test set. We train the knowledge retriever with the Adam optimizer (Kingma & Ba, 2015). The number of training epochs is set as 500 with the batch size as 100, and we use the early stopping strategy with a patience of 5. The learning rate is set as 1e-3 and the negative sampling number is set as 20. The margin $\gamma$ is also set as 5. Note that we only train the parameters within MLPs and the embeddings of items in the protein knowledge graph are frozen, thus making our knowledge retriever seamlessly generalize to knowledge updates. During inference, we rank all the candidate knowledge for a new protein based on their scores $\mathbb{S}$ (lower is better), and then select top-$K$ knowledge to add to the protein knowledge graph, where $K \in \{1,5,50,100\}$.

**Fine-tuning Implementation Details.** In the fine-tuning stage, we freeze the knowledge projector $MLP_{knowledge}$ and the structure projector $MLP_{structure}$, and only optimize the parameters within the protein encoder for downstream tasks. Note that the protein-protein interaction identification, the protein-protein binding affinity prediction, and the semantic similarity inference tasks do not need fine-tuning and we directly use the pre-trained Kara to encode proteins for these tasks. For the structure-based regularization term, we still set the margin $\gamma$ as 5 and the number of negative samples as 2. Different downstream tasks require various fine-tuning hyper-parameters and we summarize them in Table 12. Additionally, we follow the implementations in GNN-PPI (Lv et al., 2021) for PPI prediction, where the number of epochs is 600 and batch size is 2048. The learning rate is set as 1e-3 for the SHS27K dataset and 1e-4 for the SHS148K and STRING datasets. We follow the implementations in PROBE (Unsal et al., 2022) for binding affinity prediction and semantic similarity inference.

## F. Generalization Ability of Knowledge Retriever

To evaluate the generalization ability of the knowledge retriever on unseen knowledge, we employ a new data-splitting strategy. First, we randomly divide the triples (*i.e.,* (protein, relation, go)) into training and testing sets in an 8:2 ratio. Next, we remove any triple $(p_i, r_i, go_i)$ from the training set if $go_i$ appears in any test triples. This ensures that the knowledge (*e.g.,* gene descriptions) associated with test proteins is entirely absent from the training set, and thus unlearnable during training. This splitting method simulates under-studied knowledge that has gene descriptions not been observed before. The results presented in Table 13 demonstrate that our knowledge retriever can generalize to these proteins. Additionally, fine-tuning the last three layers of the PubMedBert encoder during training further improves its performance, highlighting its potential to generalize to unseen gene descriptions through domain-specific fine-tuning.

