# OpenReview forum: "Retrieval-Augmented Language Model for Knowledge-aware Protein Encoding"
_ICML.cc/2025/Conference — ICML 2025 poster_

### Official Review · Reviewer_muG4 · 2025-03-09

**Overall Recommendation:** 3

**Summary:**

The paper presents Kara, a knowledge-aware retrieval-augmented protein language model, designed to explicitly integrate knowledge from protein knowledge graphs (PKGs) into protein language models (PLMs). Unlike previous methods that implicitly embed knowledge, Kara directly injects structured knowledge through contextualized virtual tokens, allowing seamless knowledge integration during both pre-training and fine-tuning. A knowledge retriever dynamically retrieves gene descriptions for new proteins, ensuring the model continuously adapts to knowledge updates. Extensive experiments across six protein-related tasks demonstrate that Kara consistently outperforms existing knowledge-enhanced models by effectively capturing high-order biological relationships.

## update after rebuttal

The author responses address most of my concerns. I will keep my positive rating.

**Claims And Evidence:**

Overall, the paper's main claims are mostly supported by experimental evidence, but some key assumptions require further verification.
Well-supported claims:
1. Kara outperforms existing knowledge-enhanced PLMs (e.g., KeAP, OntoProtein) across six protein-related tasks. The paper provides comprehensive evaluations and ablation studies showing consistent improvements.
2. Kara effectively mitigates catastrophic forgetting by integrating knowledge into both pre-training and fine-tuning. The use of a knowledge retriever ensures continuous knowledge updates, which static models lack.
3. Explicit knowledge injection and high-order structure modeling enhance PLM performance. The introduction of contextualized virtual tokens improves knowledge integration and retrieval efficiency.
Claims needing stronger justification:
1. The advantage of direct knowledge injection over implicit embedding is not clearly isolated. A controlled experiment comparing models with virtual tokens but without retrieval is needed to confirm its impact.
2. This paper lacks visualization or biological case studies to demonstrate how Kara's representations align with real-world protein functions.

**Essential References Not Discussed:**

I have already mentioned the missing related research in the previous question “Relation To Broader Scientific Literature”.

**Experimental Designs Or Analyses:**

Strengths of experimental designs/analysis:
1. The six protein-related tasks cover both structural and functional modeling, ensuring practical relevance.
2. Ablation studies show that removing virtual tokens, retrieval, or structure-based regularization leads to performance drops, proving their importance.
Limitations:
1. The study does not evaluate Kara on low-identity or novel proteins, making its generalization unclear.
2. Only ProtBert is tested as the encoder, without comparisons to models like ESM-1b, limiting generalizability.

**Methods And Evaluation Criteria:**

The proposed methods are well-designed for integrating knowledge into protein language models and offer several unique advantages.
1. Unlike previous models that only inject knowledge during pre-training, Kara ensures continuous knowledge updates via retrieval, preventing catastrophic forgetting.
2. Contextualized virtual tokens enable direct knowledge and structure fusion. By representing gene ontology (GO) annotations and functionally similar proteins as learnable tokens, Kara effectively integrates biological insights at the sequence level.
3. Knowledge retriever dynamically aligns new proteins with the knowledge graph. Instead of relying on static embeddings, Kara retrieves relevant gene descriptions for unseen proteins, improving generalization.
4. Six tasks are well-chosen to assess Kara’s ability to model sequence-function relationships, demonstrating its applicability to real-world biological challenges.

**Other Comments Or Suggestions:**

See above.

**Other Strengths And Weaknesses:**

Kara improves protein language models with structured knowledge, making it highly relevant to biological AI. It creatively integrates retrieval, virtual tokens, and structure-based regularization to enhance knowledge utilization. The paper is well-structured and clearly explains the methods, experiments, and results.
However, the approach closely follows retrieval-based NLP models and does not introduce fundamentally new ML architectures. The paper also does not discuss how Kara performs on novel proteins with missing or incomplete knowledge.

**Questions For Authors:**

Please refer to the previous sections for key points regarding Claims and Evidence, Methods and Evaluation Criteria, Experimental Designs or Analyses, etc.

**Relation To Broader Scientific Literature:**

This paper builds on prior work in protein language modeling and knowledge-enhanced machine learning. It follows models like OntoProtein and KeAP, which integrate protein knowledge graphs (PKGs) into language models, but differs by introducing explicit knowledge injection through contextualized virtual tokens and a knowledge retriever. This aligns with trends in retrieval-augmented language models, similar to approaches in retrieval-based NLP models that dynamically fetch external knowledge. The use of structure-based regularization is also inspired by contrastive learning methods, commonly used in representation learning to enforce semantic consistency. While Kara advances knowledge integration in PLMs, it does not explore multimodal protein representations (e.g., 3D structures), which have gained traction in recent biological AI research.

**Theoretical Claims:**

The paper primarily focuses on methodological innovations rather than formal theoretical derivations. The key claims rely on empirical results rather than mathematical proofs. The loss functions and optimization strategies, such as structure-based regularization, are well-defined and align with standard machine learning principles.

---

> ### Author Rebuttal · Authors · 2025-03-31
>
> __Thanks for your kind comments, we place all tables and figures in this anonymous link(https://anonymous.4open.science/r/Rebuttal-F1C0/README.md)__
>
> ``W1. Claims needing stronger justification: (1) The advantage of direct knowledge injection over implicit embedding is not clearly isolated. A controlled experiment comparing models with virtual tokens but without retrieval is needed to confirm its impact.
> (2) This paper lacks visualization or biological case studies to demonstrate how Kara's representations align with real-world protein functions.``
>
> (1) Thanks for your kind comment. Here, we present the performance of different models for proteins within knowledge graphs. For Kara with virtual tokens but without retrieval, we construct virtual tokens using the ground-truth knowledge of each protein from the knowledge graph. In contrast, the original Kara employs a retriever to predict relevant knowledge for virtual token construction.
>
> As shown in Table 10, the variant without retrieval outperforms KeAP and OntoProtein, which embed knowledge implicitly, demonstrating the advantage of direct knowledge injection. Furthermore, the original Kara achieves performance comparable to the variant without retrieval, which utilizes ground-truth knowledge. This result highlights the effectiveness of the proposed retriever in accurately predicting protein knowledge.
>
> (2) We have provided a visualization case study comparing Kara and KeAP on the contact prediction task in "case_study_Figures.png". The results indicate that Kara outperforms KeAP in predicting contacts for proteins with short sequences (e.g., cases 1, 4, 5, and 7). However, as the sequence length increases, both Kara and KeAP struggle to accurately align with the ground truth contact map (e.g., cases 2, 3, and 6). This limitation may stem from the lack of protein structural information modeling, which is crucial for effectively handling long-sequence proteins.
>
> ``W2. The study does not evaluate Kara on low-identity or novel proteins, making its generalization unclear.``
>
> We would like to clarify, as stated in Appendix B (Lines 637–639), that we removed all proteins appearing in the downstream task datasets from the protein knowledge graph. Consequently, during inference, all proteins in the downstream tasks were unseen during training, and no related knowledge existed in the knowledge graph. Therefore, all results presented in the original paper reflect the model's performance on unseen proteins. The significant advantage of Kara over existing models demonstrates its strong generalization ability to unseen proteins.
>
> ``W3. Only ProtBert is tested as the encoder, without comparisons to models like ESM-1b, limiting generalizability.``
>
> We would like to clarify that Table 7 in the original paper already presents the performance of Kara using different encoders, including ProtBert, ProteinBert, and ESM-1b.
>
> ``W4. The approach closely follows retrieval-based NLP models and does not introduce fundamentally new ML architectures.``
>
> Thanks for your kind comment. As we have discussed in Appendix C (Lines 643-668), there are several key differences between our model and the retrieval-based NLP models, highlighting our model as an enhanced paradigm for integrating protein knowledge graphs into protein language models.
>
> (1) NLP approaches using virtual tokens assume that all encoding objectives exist in a knowledge graph, allowing direct extraction of relevant information. However, this assumption fails in protein encoding, where many proteins are absent from KGs. Our model addresses this by introducing a knowledge retriever that predicts gene descriptions for unseen proteins, enabling generalization beyond predefined KG entities.
>
> (2) Existing retriever-based NLP models use general KGs but cannot account for the unique complexities of protein KGs. Protein KGs contain multi-modal entity types requiring specialized retrieval mechanisms, and they contain large and complex gene descriptions, making the retrieval time-consuming. Our model overcomes these challenges through multi-modal matching loss and relation-go combination strategies.
>
> (3) Previous retriever-based NLP models primarily target document encoding, where KG entities are words within text corpora. These methods fail in protein encoding, since both the encoding objective and KG entities are protein sequences. Moreover, they only incorporate one-hop neighbors, overlooking higher-order structural relevance critical for protein functionality. Our model incorporates structure-based regularizations to address these limitations.

---

### Official Review · Reviewer_VkW4 · 2025-03-10

**Overall Recommendation:** 2

**Summary:**

This article proposes a knowledge-aware retrieval-augmented protein language model named Kara. During the pre-training phase, it extracts structural and knowledge information from protein KGs through contextualized virtual tokens, which are jointly embedded into the protein sequence encoding. The optimization objectives of both the pre-training and fine-tuning stages are unified through structure-based regularization. In the fine-tuning stage, a knowledge retriever predicts potential GO representations for new proteins within the PKG, thereby alleviating the issue of catastrophic forgetting of knowledge that has plagued previous models. Across multiple downstream tasks in protein prediction, Kara surpasses the current state-of-the-art models.

**Claims And Evidence:**

Yes. The experimental design and ablation study validated the effectiveness of the model and the necessity of each structure.

**Essential References Not Discussed:**

The technical elements in this paper are easy to follow, and no more references are in need to make the paper more readable.

**Experimental Designs Or Analyses:**

Yes. The article conducts comprehensive experiments on multiple tasks of protein prediction. It performs three independent experiments and takes the average to avoid the impact of randomness. Data partitioning also adheres to previous works, such as removing overlapping proteins between the training and test sets, ensuring the accuracy of the experiments. The results reported in the tables and figures can be more cafully compared to state-of-the-art.

**Methods And Evaluation Criteria:**

The six experimental datasets and Tape benchmark mentioned in the article are commonly used prediction datasets in the field of protein language modeling. Protein language models have practical significance for predicting the properties and functions of newly discovered proteins. However, the baselines used in the paper should include more recent papers.

**Other Comments Or Suggestions:**

No.

**Other Strengths And Weaknesses:**

Firstly, it introduces a dynamic retrieval mechanism from general language models into protein language models, supporting the dynamic updates of knowledge graphs, which is suitable for the rapid iteration needs in the biomedical field. The main issue is that combining retrieval information with sequence encoding lacks innovation and does not explore deeper graph structures.

**Questions For Authors:**

Have you considered applying retrieval augmentation directly to the inference stage instead of using the retriever only in the fine-tuning phase?

**Relation To Broader Scientific Literature:**

The article extensively cites relevant protein models (such as ProtBert, ESM) and knowledge-enhanced methods (including OntoProtein, KeAP), and provides detailed experimental comparisons.

**Theoretical Claims:**

There is no theoretical claims.

---

> ### Author Rebuttal · Authors · 2025-03-31
>
> __Thanks for your kind comments, we place all tables and figures in this anonymous link(https://anonymous.4open.science/r/Rebuttal-F1C0/README.md)__
>
> ``W1. The baselines used in the paper should include more recent papers.``
>
> Thanks for your kind comment. As shown in Table 1, we compare the performance of Kara with more recent and powerful models, SaProt [1] and ProSST [2], on the ProteinGYM benchmark. Kara, with the ESM2 backbone, performs competitively with SaProt but falls short of ProSST due to the lack of structural information. In the inductive setting, where test proteins are unseen and no structural or prior knowledge is available, Kara outperforms both models (Table 2). This is because Kara can automatically align unseen proteins with the protein knowledge graph through the retriever, thus enhancing generalization. In contrast, SaProt and ProSST perform poorly as they heavily rely on manually curated protein structures.
>
> [1] ProSST: Protein Language Modeling with Quantized Structure and Disentangled Attention. NuerIPS 2024
>
> [2] SaProt: Protein Language Modeling with Structure-aware Vocabulary. ICLR 2024
>
> ``W2. Combining retrieval information with sequence encoding lacks innovation and does not explore deeper graph structures.``
>
> Thanks for your kind comment. Firstly, our model is not a simple combination of retrieval information with sequence encoding, it aims to address three key technical problems for knowledge-aware protein encoding:
>
> (1) KGs are consistently updated in the real world, how to avoid the model using outdated knowledge?
>
> (2) Many newly observed proteins are understudied and thus do not exist in KG. How can we generalize the model to these understudied proteins?
>
> (3) Usually, we need to fine-tune the model to adapt to various downstream applications. How can we ensure the knowledge learned during pre-training is not to be catastrophically forgotten during fine-tuning?
>
> To address these problems, many protein modeling-specific challenges arise, such as how to align multiple modalities of information in protein knowledge graphs, how to scale the retriever to large-scale protein knowledge graphs, and how to unify the knowledge pre-training and task-oriented fine-tuning optimization objectives. The proposed Kara is incorporated with new methodologies designed especially to address these challenges. We propose a knowledge retriever to solve the modality alignment and scaleability challenges through multi-modal matching loss and relation-go combination strategies, and the contextualized virtual tokens with structure-based regularization are proposed to unify knowledge modeling during pre-training and fine-tuning.
>
> In summary, Kara contains several unique technical designs (e.g., knowledge retriever and structure-based regularizations) to solve special challenges in protein encoding scenarios, making it different from previous methods.
>
> Second, our model has incorporated both first-order and second-order graph structures, as these local structures contain the most informative knowledge for a protein. We agree that exploring more complex graph structures is an interesting direction, but given the increased retrieval complexity, we tend to consider this in future research.
>
> ``W3. Have you considered applying retrieval augmentation directly to the inference stage instead of using the retriever only in the fine-tuning phase?``
>
> We have to clarify that the proposed Kara does not use the retriever only in the fine-tuning phase, instead, it uses the retriever both during fine-tuning and inference. During fine-tuning, the retriever is used to align downstream proteins with the knowledge graph, thus unifying the pre-training and fine-tuning objectives. During inference, the retriever is used to predict potential knowledge descriptions for unseen proteins, and thus explicitly integrate knowledge for protein encoding. Table 4 presents the model’s performance when the retriever is removed during fine-tuning or inference. The results show a substantial performance drop of Kara in the absence of the retriever, highlighting the critical role of the retriever in the effective knowledge integration.

---

### Official Review · Reviewer_pk6Q · 2025-03-12

**Overall Recommendation:** 3

**Summary:**

This paper proposes Kara, a knowledge-aware retrieval-augmented language model for protein representation learning, explicitly integrating protein knowledge graphs (PKGs) with protein language models (PLMs). The key innovation lies in using contextualized virtual tokens and a knowledge retriever, allowing explicit integration of structured and task-specific knowledge during both pre-training and fine-tuning phases. The model demonstrates superior performance across multiple downstream protein tasks, including amino acid contact prediction, protein-protein interaction (PPI) prediction, homology detection, and protein stability prediction, surpassing existing baselines (e.g., ProtBert, OntoProtein, KeAP) with significant margins.

## Update after rebuttal

I appreciate the authors' comprehensive and thoughtful rebuttal. The additional experiments, including the case study on contact prediction, robustness evaluation on alternate knowledge graphs, and further clarification on catastrophic forgetting, were informative and addressed most of my earlier concerns.

The strategy to mitigate catastrophic forgetting—via explicit virtual token design and continual alignment through the knowledge retriever—is clearly explained and well-supported by the updated results. I also found the scalability discussion reassuring in terms of Kara’s practicality for large-scale knowledge graphs.

That said, there is still room for **deeper empirical error analysis** and **broader generalizability validation on different KGs**. These are directions that could further improve the work, but do not critically undermine the current contribution.

Overall, I keep my score of 3.

**Claims And Evidence:**

Most claims are well-supported by comprehensive evidence from the experiments:

1. The explicit integration of knowledge graphs into PLMs significantly enhances downstream protein representation tasks (see Tables 1, 2, and 3).

2. Contextualized virtual tokens effectively incorporate graph structures and gene ontology information, as demonstrated by the ablation studies (refer to Table 6).

However, the following claim requires further clarification:

- Claim: Kara effectively avoids catastrophic forgetting due to the unified integration of knowledge across training stages.

This claim is partially supported, as shown in Table 2 and Table 8. However, more insights and case studies are needed to illustrate scenarios of catastrophic forgetting. What is the formal definition of catastrophic forgetting? Does it refer to the decline in performance of a Protein LM when fine-tuned on a large number of data?

**Essential References Not Discussed:**

I am not aware of any essential references being missing. I think the author did a great job in Appendix C discussing the differences compared to retrieval-augmented LMs in other fields.

**Experimental Designs Or Analyses:**

**Strengths:**

1. A robust experimental design that includes multiple relevant downstream tasks and thorough comparisons with various competitive baselines.

2. It features ablation studies that clearly illustrate the contributions of different components in the model.

**Weaknesses:**
1. There is a limited examination of how sensitivity varies based on the quality or completeness of the knowledge graph. Empirical studies that specifically analyze the effects of knowledge graph incompleteness or noise are lacking. It would be beneficial to test a different protein knowledge graph to determine if the proposed method is generalizable.

2. There is a lack of comprehensive error analysis, particularly in situations where Kara underperforms or fails.

**Methods And Evaluation Criteria:**

The proposed methods and evaluation criteria align well with standard practices in protein representation learning.

1. **Methodology**:
   - Clearly detailed, integrating knowledge via contextualized virtual tokens.
   - Incorporates structure-based regularization (Section 3) is appropriate.

2. **Evaluation Criteria**:
   - Includes different metrics for each downstream tasks, which are comprehensive.
   - Choice of downstream evaluation tasks is suitable and diverse.

**Other Comments Or Suggestions:**

1. Propose a deeper analysis of robustness against incomplete or noisy PKGs, including:
   - Edge dropout experiments
   - Perturbation experiments

2. Recommend the following for real-world deployment:
   - Exploration of computational overhead
   - Scalability analysis

**Other Strengths And Weaknesses:**

**Strengths:**
- Methodologically innovative, with a clear integration of knowledge graphs and PLMs.
- Strong empirical validation showing consistent improvements across a variety of downstream tasks.
- Clearly communicates the methodological advantages and potential real-world biological implications.

**Weakness:**
- The discussion does not address computational overhead and the practical scalability of very large-scale protein knowledge graphs.

**Questions For Authors:**

If the authors could provide insights into the questions raised in the previous sections and the below questions, I would be happy to consider raising the score.
1. Could you provide explicit evidence or case studies on scenarios of catastrophic forgetting, as well as more insights on how Kara mitigates these issues?
2. How sensitive is Kara to errors or noise in the predictions made by the knowledge retriever in the finetuning stage?
3. Have you tested Kara's robustness with varying qualities of textual descriptions associated with GO entities? How does performance change based on the quality of these descriptions?
4. Can you discuss the practical scalability of very large-scale protein knowledge graphs?
5. Could you provide a case study for error analysis, particularly when Kara underperforms or encounters failures?

**Relation To Broader Scientific Literature:**

The proposed method builds upon existing literature, situating itself within knowledge-enhanced protein language modeling (e.g., OntoProtein, KeAP). It explicitly extends prior work by incorporating structured, task-oriented knowledge into the fine-tuning and inference phases of PLM.

**Theoretical Claims:**

The paper makes no explicit theoretical claims, thus no theoretical evaluation is required.

---

> ### Author Rebuttal · Authors · 2025-03-31
>
> __Thanks for your kind comments, we place all tables and figures in this anonymous link(https://anonymous.4open.science/r/Rebuttal-F1C0/README.md)__
>
> ``W1. The following claim requires further clarification: Kara effectively avoids catastrophic forgetting due to the unified integration of knowledge across training stages.``
>
> We would like to clarify that "catastrophic forgetting" in this context refers to the loss of protein attribute knowledge learned during pretraining on the protein knowledge graph due to parameter updates during downstream fine-tuning.
>
> To evaluate Kara’s effectiveness in mitigating catastrophic forgetting, we designed two experiments. The first measures the similarity between the embeddings of two proteins with the same attribute knowledge—a higher cosine similarity indicates better retention of knowledge information. The second requires the model to identify, from a set of candidate proteins, the one sharing attribute knowledge with a given protein. Higher accuracy suggests better embedding and preservation of knowledge information.
>
> As shown in Table 5, OntoProtein, Keap, and Kara all perform well after pretraining, confirming their ability to learn attribute knowledge. Kara achieves the highest performance, demonstrating its superior knowledge acquisition capability. After fine-tuning on downstream tasks, Kara’s performance remains stable, whereas OntoProtein and Keap show significant drops, indicating that they lose some of the knowledge acquired during pretraining. Furthermore, removing the structure loss or virtual token leads to performance degradation after fine-tuning, highlighting the importance of unified knowledge integration in mitigating catastrophic forgetting.
>
> ``W2. There is a limited examination of how sensitivity varies based on the quality or completeness of the knowledge graph.``
>
> First, we have to clarify that Table 8 in the original paper already analyzes the performance of different models when dealing with incomplete knowledge graphs (Edge dropout experiments). For retrieval noises analysis (Perturbation experiments), please refer to Reviewer rtvB-Q2. In Table 9, we provide the performance of Kara with ProteinKG65, showing its generalization ability to different knowledge graphs.
>
> ``W3. There is a lack of comprehensive error analysis, particularly in situations where Kara underperforms or fails.``
>
> Thank you for your thoughtful comment. We have provided a visualization case study comparing Kara and KeAP on the contact prediction task in "case_study_Figures.png". The results indicate that Kara outperforms KeAP in predicting contacts for proteins with short sequences (e.g., cases 1, 4, 5, and 7). However, as the sequence length increases, both Kara and KeAP struggle to accurately align with the ground truth contact map (e.g., cases 2, 3, and 6). This limitation may stem from the lack of protein structural information modeling, which is crucial for effectively handling long-sequence proteins.
>
>
> ``W4. Computational overhead and the practical scalability of very large-scale protein knowledge graphs``
>
> As discussed in lines 263–274 on page 5, we designed the relation-GO combinations strategy to generalize to large-scale KGs. Table 6 presents the time cost of Kara during training and inference with and without the retriever. The results show that, after applying the relation-GO combinations strategy, Kara's training and inference time only slightly increases compared to knowledge-free baselines (even on large-scale knowledge graph ProteinKG65), demonstrating its scalability.
>
> ``Q1. More insights on how Kara mitigates catastrophic forgetting.``
>
> Existing approaches like KeAP and OntoProtein employ knowledge graph-supervised pre-training to encode knowledge information into model parameters, followed by task-specific fine-tuning using proteins from downstream tasks. However, the absence of knowledge descriptions for proteins in downstream tasks creates a knowledge supervision gap during fine-tuning. This causes the model optimization to focus solely on task objectives, potentially overwriting the previously acquired knowledge representations in parameters and leading to catastrophic knowledge forgetting.
>
> In contrast, our proposed Kara framework addresses these issues through two key innovations. First, rather than implicitly storing knowledge in model parameters, Kara explicitly incorporates knowledge through virtual tokens. This architectural design decouples knowledge storage from model parameters, making the acquired knowledge resilient to parameter updates during downstream fine-tuning. Second, Kara introduces a knowledge retriever that aligns downstream task proteins with the knowledge graph to predict potential knowledge descriptions. This alignment mechanism enables continuous knowledge supervision during fine-tuning, ensuring that parameter updates simultaneously optimize for both task performance and knowledge consistency.

---

> > ### Comment · Reviewer_pk6Q · 2025-04-05
> >
> > Thanks for your response, which addresses the majority of my concerns. I will keep the current positive score.

---

### Official Review · Reviewer_rtvB · 2025-03-13

**Overall Recommendation:** 3

**Summary:**

The paper presents Kara, a knowledge-aware retrieval-augmented protein language model that explicitly integrates protein knowledge graphs (PKGs) with protein language models (PLMs), enhancing protein representation learning with task-specific knowledge and graph structure information.
Kara predicts potential gene descriptions for proteins (via knowledge retriever), aligning them with PKGs before injecting the task-relevant high-order graph structure information into protein representations with contextualized virtual tokens.
They propose the use of structure-based regularization to maintain consistency between pre-training and fine-tuning objectives.
Kara outperforms existing models like knowledge-enhanced PLMs KeAP** and ESM-2 across six downstream tasks.
The paper proposes different ablation studies, e.g., the effectiveness of each component, numbers of knowledge, and knowledge retriever.

**Claims And Evidence:**

1. The main claim of the paper (*Kara improves protein function modeling by explicitly incorporating task-oriented knowledge*) is clear and supported by empirical evidence.
* Kara outperforms KeAP and ESM-2 across six benchmark tasks, e.g., Kara achieves 11.6$\%$ improvement in long-range contact prediction.
* Stronger baselines may be considered (though they utilize further protein structural information) like SaProt (https://openreview.net/forum?id=6MRm3G4NiU) or ProSST (https://www.biorxiv.org/content/10.1101/2024.04.15.589672v3), which shows to achieve higher performance, e.g., on contact prediction. Furthermore, it seems the ESM2 performance reported in SaProt (Table 3) for contact prediction is higher than the scores in this paper.

2. The following claims are clear and intuitive but require further evidence to be strongly supported.
- (1) The knowledge retriever enhances generalization to unseen proteins.
- (2) Kara mitigates catastrophic forgetting through unified knowledge integration.

3. The ablation studies confirm the contributions of virtual tokens, structure-based regularization, and retrieval mechanisms.
However, an error analysis on the effect of the retriever on the performance when it introduces  incorrect knowledge

**Essential References Not Discussed:**

No

**Experimental Designs Or Analyses:**

Yes, I checked all experiments for 6 downstream tasks and ablation studies.

**Methods And Evaluation Criteria:**

**Methods**
- The proposed framework is sound and intuitive.
- However, it should indicate the assumptions in which the method works: (i) Relevant knowledge for a protein exists in the PKG, (ii) Graph structure provides meaningful relationships.

**Evaluation**
- The benchmark is comprehensive (amino acid contact prediction, PPI identification, homology detection, stability prediction). However, the fitness prediction (zero-shot) for PLMs should be considered together with perplexity to validate that the model prevents forgetting.
- Ablation studies validate each component’s contribution and provide insights into retrieval behavior.
- Baselines: As mentioned above, the paper may consider the latest baselines that show higher performance on these tasks.

**Other Comments Or Suggestions:**

- The "limitations" in the introduction may cause a misunderstanding as to the limitations of this work. The writing of this part can be polished for better understanding with cited evidence.
- Some sections are very dense, e.g., 3.2 and 3.3, with some repetitive/unnecessary information/notations that can be trimmed for better readability.

**Other Strengths And Weaknesses:**

**Strengths**:
  - The problem of interest is interesting and timely.
  - The proposed method is sound and useful for further research on PLMs.
  - The paper is well-structured

**Weaknesses**:
  - Kara depends on the retriever. If the retriever selects incorrect gene descriptions, it may degrade performance rather than improve it.
  - The retrieval complexity scales better than brute-force retrieval, but it is unclear how well it generalizes to large-scale PKGs, which serves the purpose of the paper. Furthermore, how does the retrieval overhead compare to knowledge-free baselines?

**Questions For Authors:**

1. What happens when the retrieval mechanism introduces incorrect or irrelevant gene descriptions? How does noisy or incomplete PKG knowledge affect the model?

2. Can retrieval be extended to handle proteins with missing or sparse knowledge? What happens when a protein has no useful GO entities?

3. Instead of fixed K retrieved entities, could a confidence-based retrieval mechanism improve accuracy? Is there a trade-off between retrieval depth and performance?

4. Given the proteins in the PKGs for inference, can you elaborate why explicitly using knowledge (as of Kara) is better then the one encoded in pretraining?

**Relation To Broader Scientific Literature:**

- The problem of knowledge-aware protein representation learning is highly relevant to current works of protein language model and computational biology.
- The paper clearly articulates the limitations of previous methods and proposes a novel retrieval-based augmentation.

**Theoretical Claims:**

No theoretical claim.

---

> ### Author Rebuttal · Authors · 2025-03-31
>
> __Thanks for your kind comments, we place all tables in this anonymous link(https://anonymous.4open.science/r/Rebuttal-F1C0/README.md)__
>
> ``W1. Stronger baselines may be considered.``
>
>  As shown in Table 1, we compare the performance of Kara, SaProt, and ProSST on the ProteinGYM benchmark. Kara, with the ESM2 backbone, performs competitively with SaProt but falls short of ProSST due to the lack of structural information. In the inductive setting, where test proteins are unseen and no structural or prior knowledge is available, Kara outperforms both models (Table 2). This is because Kara can automatically align unseen proteins with the protein knowledge graph through the retriever, thus enhancing generalization.
>
> ``W2. ESM2 performance reported in SaProt for contact prediction is higher.``
>
> They used the 33-layer ESM2-33t model, while we opted for the 30-layer ESM2-30t model to ensure a fair comparison with previous knowledge graph-based models, which used a 30-layer ProtBert as the backbone. Despite having fewer parameters and layers than ESM2-33t, our model outperforms ESM2-33t on the short-term Contact task (Table 3), and achieves comparable performance on the ProteinGYM benchmark, as shown in Table 1.
>
> ``W3.  Claims require further evidence. (1) The knowledge retriever enhances generalization to unseen proteins. (2) Kara mitigates catastrophic forgetting through unified knowledge integration.``
>
> (1) As stated in Appendix B (Lines 637–639), we removed all proteins appearing in the downstream task datasets from the protein knowledge graph. Consequently, during inference, all proteins in the downstream tasks were unseen during training, and no related knowledge existed in the knowledge graph. Therefore, all results presented in the original paper reflect the model's performance on unseen proteins.
>
> Additionally, Table 4 presents the model’s performance when the retriever is removed during fine-tuning and inference. The results show a substantial performance drop in the absence of the retriever, further highlighting its critical role in enhancing generalization.
>
> (2) Due to space limitations, please refer to Reviewer pk6Q-W1.
>
> ``Q1. How well does Kara generalize to large-scale PKGs.``
>
> As discussed in lines 263–274 on page 5, we designed the relation-GO combinations strategy to generalize to large-scale KGs. Table 6 presents the time cost with and without the retriever. The results show that, after applying this strategy, Kara's training and inference time only slightly increases compared to knowledge-free baselines (even on large-scale ProteinKG 65), demonstrating its scalability.
>
> ``Q2. How does retrieval noise or incomplete PKG affect the model.``
>
> Table 8 in the original paper already analyzes the performance of different models when dealing with incomplete knowledge graphs.
>
> Table 7 presents the performance when varying levels of noise are introduced into the retrieved results (by replacing retrieved knowledge with random knowledge). The results indicate that retrieval noise does not significantly impact performance. This is because the model's fine-tuning process does not rely on ground-truth knowledge and is inherently noisy, enhancing the model's robustness.
>
> ``Q3. Can retrieval be extended to handle proteins with missing knowledge?``
>
> Our retriever is inherently designed to handle unseen proteins without relevant prior knowledge. It achieves this by predicting the most likely knowledge description for an unseen protein, linking it to the KG, and retrieving relevant structure as knowledge information. Therefore, regardless of whether a protein has associated knowledge or useful GO entities, the retriever can still process it. While prediction errors may occur, the impact of retrieval noise on model performance due to such errors has already been discussed in Q2.
>
> ``Q4. Is there a trade-off between retrieval depth and performance? Could a confidence-based retrieval mechanism improve accuracy?``
>
> We propose a relation-GO combinations strategy, which balances the breadth and performance of retrieval by identifying the interacting entities specific to each relation.  Moreover, the retrieval method does not require deep-first graph search, making it scalable for large KGs, as discussed in response to Q1.  The performance of using the confidence threshold is provided in Table 8, showcasing a slight performance improvement.
>
> ``Q5. Why explicitly using knowledge (as of Kara) is better than the one encoded in pretraining?``
>
> First, previous works have demonstrated that encoding knowledge directly into parameters can not accurately retain the knowledge information [1]. Second, KGs are subject to updates. Models based on parameter encoding cannot easily adapt to such updates and require retraining. In contrast, Kara explicitly integrates knowledge through virtual tokens, enabling it to seamlessly adapt to updates in the knowledge graph.
>
> [1] Large language models struggle to learn long-tail knowledge. ICML 2023

---

> > ### Comment · Reviewer_rtvB · 2025-04-07
> >
> > Thank you for your clarification and update.
> > I keep my positive recommendation. Good luck.

---

### Decision · Program_Chairs · 2025-05-01

**Decision:**

Accept (poster)

**Comment:**

The paper introduces Kara, a knowledge-aware retrieval-augmented protein language model that explicitly integrates protein knowledge graphs (PKGs) with protein language models (PLMs). This integration enhances protein representation learning by incorporating task-specific knowledge and leveraging the structural information inherent in protein knowledge graphs. Experimental validations are conducted across multiple tasks to verify the effectiveness of this approach.

During the author-reviewer discussion, KeAP was thoroughly discussed, and most reviewers were convinced of its merit, although the novelty is not particularly significant. The paper is well-executed overall. Therefore, I recommend a weak accept